# Decoding the Enigma: Benchmarking Humans and AIs on the Many Facets of Working Memory

Ankur Sikarwar  and Mengmi Zhang

Deep NeuroCognition Lab, Institute for Infocomm Research (I2R), A*STAR, Singapore
Center for Frontier AI Research (CFAR), Agency for Science, Technology, and Research (A*STAR), Singapore
School of Computer Science and Engineering, Nanyang Technological University, Singapore
Address correspondence to mengmi@i2r.a-star.edu.sg

## Abstract

Working memory (WM), a fundamental cognitive process facilitating the temporary storage, integration, manipulation, and retrieval of information, plays a vital role in reasoning and decision-making tasks. Robust benchmark datasets that capture the multifaceted nature of WM are crucial for the effective development and evaluation of AI WM models. Here, we introduce a comprehensive **Wor**king **M**emory (**WorM**) benchmark dataset for this purpose. WorM comprises 10 tasks and a total of 1 million trials, assessing 4 functionalities, 3 domains, and 11 behavioral and neural characteristics of WM. We jointly trained and tested state-of-the-art recurrent neural networks and transformers on all these tasks. For comparison, we also include human behavioral benchmarks. Note that all computational models were never trained with any human data or behavioral biases; yet, these models remarkably replicate some characteristics of WM in biological brains, such as primacy and recency effects. Moreover, we performed neural population analysis on these models and identified neural clusters specialized for different domains and functionalities of WM. Not all computational models exhibit a strong alignment with all human behaviors. Our experimental results also reveal several limitations in existing models to match with working memory capabilities of humans. This dataset serves as a valuable resource for communities in cognitive psychology, neuroscience, and AI, offering a standardized framework to compare and enhance WM models, investigate WM's neural underpinnings, and develop WM models with human-like capabilities. Our source code and data are available at: link.

## 1   Introduction

Working memory (WM) defines a core cognitive process enabling temporary storage, integration, manipulation, and recall of information. It is vital for many downstream tasks involving reasoning [57, 46, 34], decision-making [50, 26], and language comprehension [5, 9, 1]. Understanding and modeling WM has significant implications for both neuroscience and AI research. In neuroscience and cognitive science, extensive investigations have been conducted to unravel the underlying mechanisms of WM. The non-exhaustive list of works involves (1) identifying brain regions such as the prefrontal cortex [15, 51, 37], hippocampus [6, 55], and parietal cortex [28, 32, 47] as key players in WM processes; (2) exploring the role of WM capacity in cognitive processes such as selective attention [17, 48], and executive control [19, 7]; and (3) investigating how WM abilities differed in individuals of different ages [20, 53], intelligence levels [12], and neurological disorders [2, 23].

However, despite significant progress in individual subdomains, there remains a notable gap in the broader study of WM and its relevance for the AI community. To date, research efforts in neuroscience, and cognitive science have mostly focused on specific aspects of WM or individual

37th Conference on Neural Information Processing Systems (NeurIPS 2023) Track on Datasets and Benchmarks.

memory tasks. There is a lack of systematic, integrative, and quantitative exploration that covers multiple tasks, encompasses various functionalities and domains of WM, and provides a systematic benchmark for evaluating both humans and AI models in these memory tasks.

In this work, we aim to address this gap by establishing a general framework to study WM. To capture the multifaceted nature of WM due to its complexity, flexibility, and variability, we include 10 WM tasks and curate 1 million trials. Each of the state-of-the-art recurrent neural networks and transformer models is jointly trained and tested on these tasks with the following goals: (1) to assess model behaviors, such as set size effect and retention interval; (2) to investigate neural populations, such as task-specialized neural clusters and neural correlates of executive control; and (3) to examine and compare performance across multiple WM models when performing different memory functions: storage, integration, manipulation, and supervision. We include human performance on the WM tasks for behavioral comparison with AI models. In the experiments, we observe that AI models replicate some human-like characteristics of WM; however, we also identify several limitations in the existing WM models, such as disparities between human and model behaviors and inferior model performances in tasks that humans excel at. Through this interdisciplinary exploration, we strive to enhance our understanding of WM processes, uncover novel insights into the cognitive architecture of human WM, and propel the development of AI systems with more robust and human-like WM capabilities.

Main contributions are highlighted:

**1.** We introduce WorM, a comprehensive large-scale WM benchmark dataset, covering 10 WM tasks and containing 1 million trials, encompassing 4 functionalities and 3 domains of WM.

**2.** We introduce evaluation metrics for all our tasks and establish a general methodology to jointly train, study, and benchmark WM models on all tasks.

**3.** We explore 8 WM behavioral benchmarks and find that recurrent neural networks are more closely aligned with humans, while transformers exhibit limitations in replicating human behavior.

**4.** We examine the neural population of WM models. Our analysis reveals task-specialized neural clusters and uncovers neural correlates of executive control in WM models.

## 2 Related Works

Working memory (WM) has been a topic of interest in cognitive psychology and neuroscience for decades. Numerous studies [40, 24, 39, 35, 61, 3, 8] have investigated different behavioral effects and their underlying mechanisms and neural correlates. There has also been an interest [14, 42, 36, 43] in developing computational models of WM that can explain and approximate human behavior on WM tasks. Several WM tasks [16, 54, 13, 35, 41, 31] have been proposed to evaluate WM performance of humans. For example, the N-back task is a widely used WM task that requires participants to remember a sequence of stimuli and respond when a stimulus matches the one presented N trials earlier. It is important to emphasize that the majority of these prior works have typically focused on isolated aspects of working memory or evaluated a single working memory model with a restricted range of working memory tasks. In contrast, our contribution involves the development of a comprehensive and quantitative benchmark that encompasses 10 distinct working memory tasks, spanning four functionalities and three domains of working memory, for the evaluation of both human performance and AI models.

In the field of AI, WM has also attracted considerable attention. AI researchers have aimed to develop computational models that can emulate and augment human-like WM capabilities. Various memory architectures [10, 27, 25, 62, 56] have been proposed, such as neural networks with memory cells, recurrent neural networks, and memory-augmented neural networks. These models have demonstrated promising results in specific WM tasks, such as copying, sorting, and memory recalls, as well as real-world applications [22, 33], such as navigation in naturalistic environments, video recognition, and language processing.

Recently, [45] established a comprehensive working memory (WM) benchmark to aid in the development of WM theories by collecting empirical findings from previous human studies in a wide range of WM tasks. Different from their work, in this paper, we present a large-scale WM

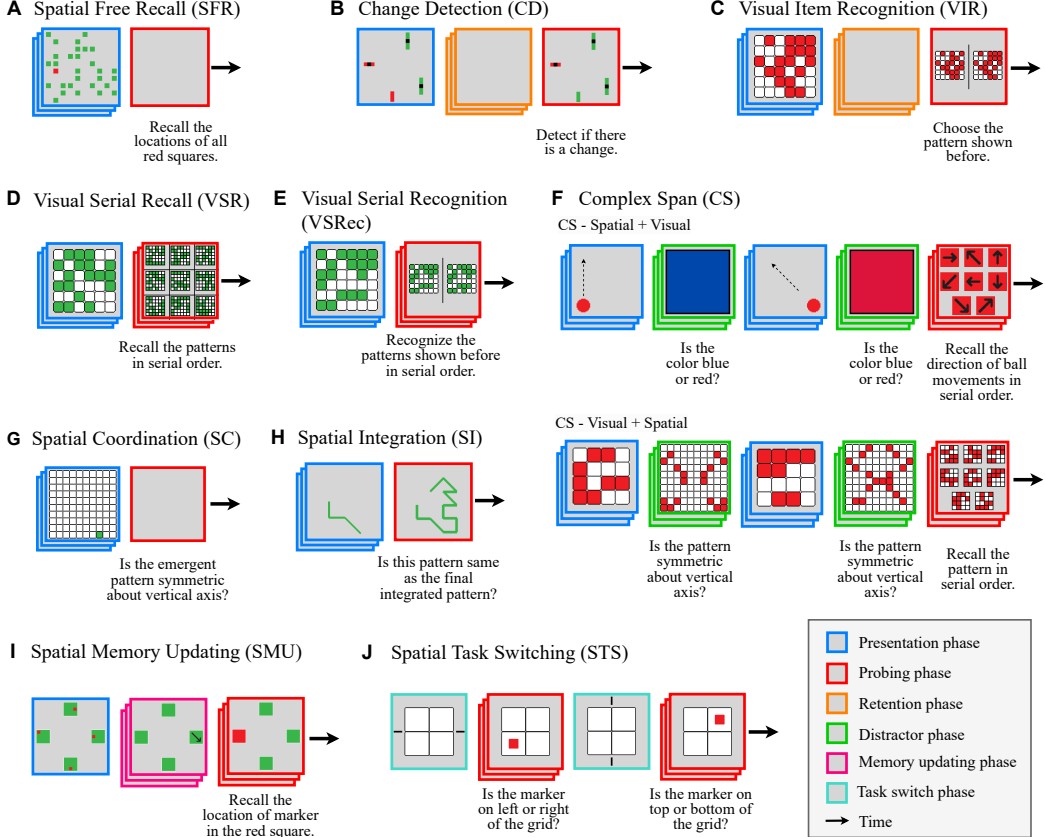

Figure 1: **Schematic illustration of all 10 working memory tasks.** Each panel illustrates the schematic for a different task. In each panel, the arrow pointing from left to right denotes the progression of time. Overlapping frames indicate multiple time steps whereas single frames represent a single time step in the trial. In F, we further divide the task into four types. Two types are shown here. See **Sec. 3** for a detailed description of each task.

benchmark dataset to bridge the gap between cognitive psychology and the AI field, allowing modern AI models to be trained on such WM tasks for direct behavioral and neural comparison.

## 3 Psychophysics Experiments in WorM

**Phase definitions in WM experiments.** In this section, we refer to our 10 WM tasks as 10 experiments. A WM experiment often consists of several but not necessarily all phases (**Fig. 1**) described below: * **Presentation phase ($\mathbf{P}_{present}$)**: A sequence of stimuli to remember is presented sequentially. * **Retention phase ($\mathbf{P}_{retent}$)**: A grey blank image is presented for a period of time, during which the remembered patterns in $\mathrm{P}_{present}$ have to be maintained in the memory. * **Probing phase ($\mathbf{P}_{probe}$)**: Responses to a series of questions asking to recall certain aspects of the memory content are required. Sometimes, there will be no questions but free recalls are required. * **Distractor phase ($\mathbf{P}_{distractor}$)**: A series of other distracting tasks are presented in this phase to interfere with the original WM experiments. * **Memory updating phase ($\mathbf{P}_{update}$)**: A series of instructions to update the memory content are given. * **Task switching phase ($\mathbf{P}_{switch}$)**: A cue image is presented to signal the transition between a pair of tasks.

**Time steps in WM experiments.** The time duration of each phase in human psychophysics experiments is often measured in milliseconds (**ms**). However, computational WM models do not have the notion of time. We establish a mapping between the time step $t$ in WM models and **ms** in human psychophysics experiments for all the phases of all the experiments (see **Appendix A2.2** for mapping details). For consistency, we describe all the main text with time steps $t$. We introduce $T$ to denote the total time steps for each experiment and $T_{phase}$ to denote the total time steps of each phase, where $phase$ could be from any of the phases introduced above. For example, $T_{present} = 3$

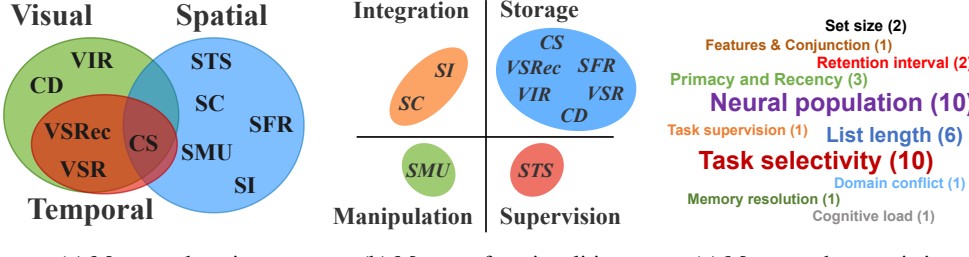

|  |  |  |
|---|---|---|
| (a) Memory domains | (b) Memory functionalities | (c) Memory characteristics |

Figure 2: **WorM covers a wide spectrum of working memory (WM) tasks encompassing 3 domains, 4 functionalities, and 11 characteristics.** For all ten tasks, we categorize them based on their objectives in studying different **(a)** memory domains, and **(b)** memory functions. See **Fig. 1** for task acronyms. In **(c)**, we present word clouds representing the studied memory characteristics. Larger fonts indicate that more tasks (the exact number in brackets) produce results in this category.

indicates that there are 3 time steps for the presentation phase and $t_{present} = 1$ indicates the first time step of the presentation phase.

**Nomenclatures in cognitive science and neuroscience.** In human psychophysics experiments [4, 13, 29], researchers often use **"length list"**, denoted as $L$, to indicate the total number of stimuli presented during each $P_{present}$, and **"serial position"** to indicate the position of the $l$th stimulus in $L$. For WM models, each stimulus is presented at each time step of $P_{present}$. Thus, $L = T_{present}$ and $t_{present} = l$ denotes the $l^{th}$ serial position during $P_{present}$. Following psychology and neuroscience literature [35], we also define **"set size"**, denoted as $S$, to be the total number of items that need to be held and manipulated in working memory at a given time step. For all the experiments, we use these terms interchangeably, and we also use "agents" to indicate that the test subjects can be both humans and computational WM models.

**Introduction to ten psychophysics experiments**

We introduce 10 psychophysics experiments on Working Memory (WM) from cognitive science and neuroscience [13, 35, 29, 4, 60, 41, 52, 44, 38]. To facilitate comparisons with computational models, we directly copy the human behavioral data collected from these existing works to our paper. Based on WM functionalities these experiments entail (**Fig. 2b**), we categorize them into four functional groups: **(A)** memory storage, **(B)** memory integration, **(C)** memory manipulation, and **(D)** memory supervision. We also show the overview of the three WM domains (visual, spatial, and temporal) that each experiment involves in **Fig. 2a**. See **Appendix A1** for experimental details.

**Experiment A1 - Spatial Free Recall (SFR) [13] (Fig. 1A).** At every $t_{present}$, a randomly chosen square on a green grid of size $10 \times 10$ turns red. In $P_{probe}$, a blank grey image is shown. The task for the agent is to remember all the previous spatial locations where the squares turned red and provide a 100-way multi-label classification response.

**Experiment A2 - Change Detection (CD) [35] (Fig. 1B).** In $P_{present}$, a memory array is shown, consisting of $S$ bars with varying attributes such as color, orientation, size, and the presence/absence of a gap in each bar. Followed by a sequence of blank grey images in $P_{retent}$, the agents are tested with a probe array in $P_{probe}$, which can be identical to the memory array or have a single bar that differs along one feature dimension. The task for the agents is to make a binary prediction about whether there is a change or not given the test array.

**Experiment A3 - Visual Item Recognition (VIR) [29] (Fig. 1C).** In $P_{present}$, a sequence of distinct $6 \times 6$ matrix patterns is displayed. In $P_{retent}$, a series of blank grey images is shown. In $P_{probe}$, a probe image is presented, consisting of two patterns side-by-side. One pattern matches a previously shown pattern, while the other is a distractor. The agents must perform a binary classification task, where 0 represents the left pattern and 1 represents the right pattern.

**Experiment A4 - Visual Serial Recall (VSR) [4] (Fig. 1D).** In $P_{present}$, a sequence of $L$ matrix patterns is sequentially presented. These patterns consist of a $6 \times 6$ grid with half of the cells filled in green. In $P_{probe}$, a probe image is displayed, containing all $L$ patterns presented during $P_{present}$. Agents perform a 9-way classification task at each $t_{probe}$, recalling the matrix patterns seen at $t_{present}$ in a serial order. See **Appendix A3.1** for a language-based variant of this experiment.

**Experiment A5 - Visual Serial Recognition (VSRec) [4]** (**Fig. 1E**). $P_{present}$ is exactly the same as $P_{present}$ in the VSR experiment. Differences were introduced during $P_{probe}$. At each $t_{probe}$, a target pattern from $P_{present}$ and a distractor pattern were presented together. Distractor patterns were distinct from the initially presented matrix patterns and differed from the target pattern by $n$ cells, where $n$ denotes the pattern-distractor difference. Agents performed a binary classification task, with 0 indicating the left pattern and 1 indicating the right pattern.

**Experiment A6 - Complex Span (CS) [60]** (**Fig. 1F**). The experiment includes two types of $P_{present}$ and $P_{probe}$, as well as two types of $P_{distractor}$. (I) The first type involves memorizing visual patterns and recalling them in order (visual). (II) The second type involves memorizing spatial patterns of ball movements and recalling their directions (spatial). Two types of $P_{distractor}$ are introduced to interfere with memory. (III) color discrimination tasks (visual) and (IV) spatial symmetry discrimination tasks (spatial) are used as distractors. Four variations of experiment conditions are considered: visual storage + visual distractor, spatial storage + visual distractor, visual storage + spatial distractor, and spatial storage + spatial distractor.

**Experiment B1 - Spatial Coordination (SC) [41]** (**Fig. 1G**). $P_{present}$ involves a $10 \times 10$ grid where one cell is highlighted in green at each $t_{present}$. In $P_{probe}$, agents are presented with a blank gray image and asked to perform a binary discrimination task to determine whether the pattern formed by integrating all the patterns in $P_{present}$ is symmetric about the vertical axis or not.

**Experiment B2 - Spatial Integration (SI) [52]** (**Fig. 1H**). $P_{present}$ involves sequentially presenting partial line drawings on a $4 \times 4$ grid, eventually completing a multi-segment figure. The number of line segments in the partial drawings determines the number of integration operations required, also equivalent to $L$. In $P_{probe}$, agents perform a binary classification task to determine if the given figure matches the mentally integrated figure from $P_{present}$.

**Experiment C1 - Spatial Memory Updating (SMU) [44]** (**Fig. 1I**). In $P_{present}$, a stimulus with green squares is presented, each containing a red marker in one of the nine possible locations on a $3 \times 3$ grid. $P_{update}$ involves arrows indicating the direction for mentally updating the marker's location. In $P_{probe}$, agents perform a 9-way classification task to recall the marker's location in the highlighted square.

**Experiment D1 - Spatial Task Switching (STS) [38]** (**Fig. 1J**). In $P_{switch}$, a cue image indicates the discrimination task for subsequent $P_{probe}$. There are two tasks across which to switch: top-versus-bottom and left-versus-right. At every $P_{probe}$, a red marker appears in a $2 \times 2$ grid, and agents switch between the two tasks based on cues presented during $P_{switch}$.

**Experimental trial splits for computational WM models.** For every WM psychophysics experiment introduced above, we generate 86,400 trials for training, 9,600 trials for validation, and 9,600 trials for testing by following a ratio of 9:1:1. These trials are distributed uniformly among various experimental conditions for individual experiments.

# 4   Computational Models of Working Memory

**Joint training and testing regime.** There exist multiple approaches to train and evaluate computational working memory (WM) models using our dataset, such as "train on many; test on one" and "train on one; test on one". However, our aim here is not to exhaustively benchmark WM models across all possible training and testing regimes. Instead, we aim to lay the foundation for a systematic and quantitative methodology to comprehensively study the multi-facets of WM. In this regard, we propose a joint learning paradigm where we simultaneously train and test different models on all conditions of all ten WM tasks. This approach allows us to explore the intricate interactions and inter-dependencies among the different WM tasks, capturing the complex nature of WM performance. For comparisons with the joint training regime, we also conduct single-task training on the VSR task. See **Appendix A3.3** for more details.

**Architectures for working memory models.** We investigate several state-of-the-art variations of recurrent neural networks, including vanilla recurrent neural networks (RNNs), Gated Recurrent Units (GRUs), and Long Short-Term Memory (LSTM) networks [10, 27]. Furthermore, considering the recent success of Transformers [59], we also include vanilla transformer-based encoders (TRF). Below, we introduce details of our model architectures (**Fig. 3**).

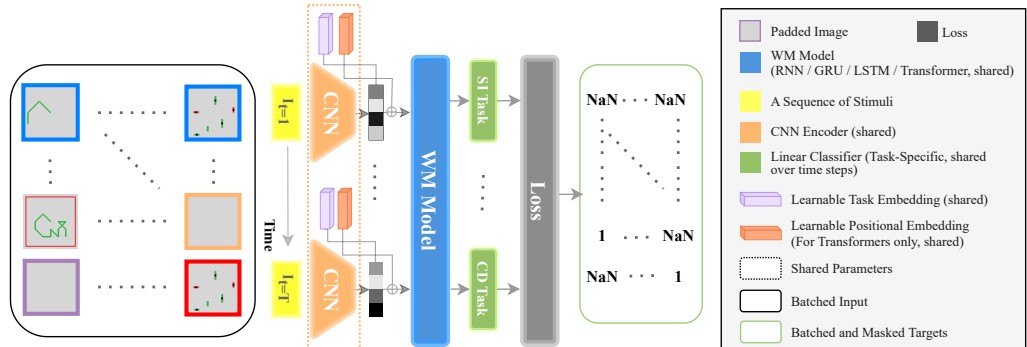

Figure 3: **Overview of Working Memory (WM) models.** All WM models take a batch of stimuli in sequences from $I_{t=1}$ to $I_{t=T}$ as inputs and predict the task-specific responses at every single time step $t$. These responses are compared against the batched and masked ground truths with classification losses. The gradients are only back-propagated based on the actual target values. For time steps without responses required (denoted as "NaN"), the predicted responses from the linear classifiers are not computed, and thus, no gradients are back-propagated. See legends for box notations. "shared" refers to learnable parameters shared jointly over tasks except for "task-specific shared" in linear classifiers, which refers to learnable weight parameters shared across time steps within the same task but different over tasks. See **Sec. 4** for model details.

**Feature Extraction.** At each time step $t$ of a trial, the visual stimulus $I_t$ comes as a $32 \times 32 \times 3$ tensor which is then fed to a 4-layer 2D-convolutional network (2D-ConvNet). See **Appendix A2.1** for the 2D-ConvNet architecture. The output feature maps from 2D-ConvNet are then flattened to a vector of dimension $K$ and linearly projected to a feature representation $F_t$ of dimension $D$ using weight matrix $\in \mathbb{R}^{K \times D}$. Different trials can be of different lengths, therefore we pad all the trials with blank images to ensure that all trials have a consistent length of 20.

**Task embeddings.** In addition to extracting stimuli features, we also introduce learnable task-specific embeddings $M \in \mathbb{R}^{14 \times 14}$, informing the network about the task identity during the joint training. At every $t$ of each task, the corresponding task embedding is concatenated with the stimulus feature representation $F_t$ from $I_t$, resulting in a (14+D)-dimensional representation $A_t$. $A_t$ is further fed to various WM models introduced below for further processing. See **Appendix A2.1** for details.

**Encoding Time-dependent Representations.** As we have RNN-based networks and transformer-based networks to process stimuli sequences, we introduce their model designs separately. For RNNs, GRUs, and LSTMs, we use only one recurrent layer. The recurrent networks take (14+D)-dimensional representation $A_t$ above as inputs for each $t$. We define the capacity of these models $C$ as the number of hidden units in this recurrent layer. At every $t$, the recurrent networks output a hidden representation $h_t$ of dimension C used for predicting task-specific responses.

For TRFs, we introduce extra learnable positional embeddings $P \in \mathbb{R}^{20 \times (14+D)}$ which is shared across all tasks. Each row of $P$ indicates a "time step". At $t^{th}$ time step, $P_t$ is incorporated into $A_t$ via element-wise addition. We denote this positional embedding-modulated stimulus feature representation as $A_{TRF,t}$ which is then fed to the TRF. We include two standard transformer encoder blocks in TRF. We define the model's capacity $C_{TRF}$ as the dimension $14 + D$ of the stimulus feature vector $A_t$. Additionally, to simulate a more realistic scenario where the TRF model can only rely on past context to make predictions, we apply masking in the self-attention mechanism to prevent the model from accessing future information. We take the output $h_{t,TRF}$ of TRF corresponding to the $t^{th}$ time step and use it for predicting task-specific responses at $t$.

**Response Generation.** The output of a WM network $h_t$ or $h_{t,TRF}$ is fed to a task-specific linear classifier with weight matrix $O_{task}^{(14+D) \times U_{task}}$ for generating final probability distributions over a fixed set of all possible response choices in the corresponding task. Note that $O_{task}$ is shared across all $T$ time steps within the same task, but it is independent over different tasks. For example, in the VSR task, during $P_{probe}$, the WM model has to predict which pattern among the 9 shown patterns is the target pattern. In this case, $U_{task} = 9$, and the linear classifier outputs a probabilistic distribution of dimension 9, and the choice with the largest probability is the final response selected by the model.

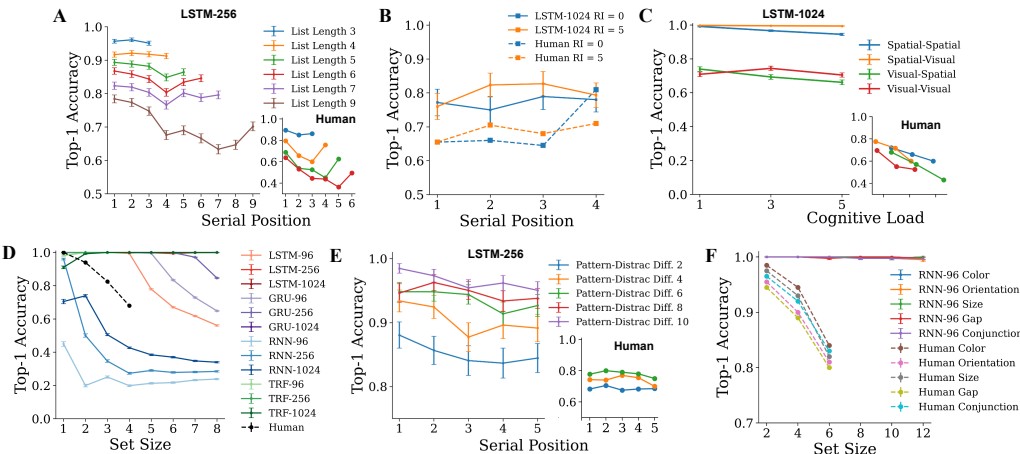

Figure 4: **Performance benchmarks and behavioral analysis for working memory models and humans.** We present the behavioral accuracy as a function of (A) list lengths $L$ for LSTM-256 in VSR task, (B) retention interval in VIR task, (C) memory domain conflicts for LSTM-1024 in CS task, (D) set sizes $S$ in SMU task, (E) memory resolution $n$ in VSRec task, and (F) number of features or conjunctions of features per item in CD task. See **Sec. 3** for all the task introductions. See **Sec. 5** for the analysis of these results.

**Training Details.** Practically, the WM models output responses for all $T$ steps. However, training supervision with corresponding losses is only applied on the set of time steps where actual responses are required. All model parameters, including the feature extractors and WM networks, are initialized using Xavier initialization and trained from scratch. See **Appendix A2.1** for more training details and **Tab. A1** for hyper-parameter configurations.

**Comparison with humans.** To quantitatively compare computational models and humans on behavioral benchmarks, we introduce two types of scores: the slope difference score ($A$) and the average accuracy difference score ($B$). We presented the experimental details and the results in **Appendix A2.3**. We conducted statistical tests on the results. We found that the majority of the models are more consistent with humans than the chance model; however, there still exists performance discrepancies among different models and performance gaps between these models and humans.

## 5   Results and Analysis

We analyze multiple memory characteristics (**Fig. 2c**) individually for the corresponding tasks for different WM models. We use "model-$C$" to refer to the WM model of memory capacity $C$. For example, LSTM-128 refers to the model with LSTM of memory capacity $C = 128$ as the backbone. As benchmarks, we copy the exact human performances here from [13, 35, 29, 4, 60, 41, 52, 44, 38] for direct comparison with WM models. Importantly, the WM models were not trained on human data or influenced by human biases. Therefore, our focus is not on comparing absolute performance disparities between humans and WM models. Instead, we aim to examine their qualitative trends across different conditions. Due to the extensive combinations of models and tasks, we present a condensed selection of results for some models in this section. See **Appendix A4** for the expanded result analysis with all model variations in these experiments. For fair comparisons among different model architectures in terms of overall WM performance, we compare different architectures with either the same memory capacity sizes $C$ or the same number of network parameters. In both comparisons, we observe that recurrent networks like LSTMs and GRUs perform better than transformers on different working memory tasks (refer **Appendix A4** for more details).

**A. Primacy and recency effect across list lengths is an emergent phenomenon.** We report the top-1 accuracy as a function of serial positions across various list lengths $L$ for LSTM-256 in the VSR task (**Sec. 3**, **Fig. 4A**). We make several observations. Firstly, as $L$ increases, the memory load in WM also increases, resulting in a decrease in overall accuracy across different list lengths (indicated by the varying colors of the lines, from blue to brown). Secondly, our findings in WM models align with the well-known cognitive science and neuroscience phenomena of the primacy and recency effects. These effects demonstrate that items presented at the beginning (primacy) and end (recency)

of a trial are better retained than those presented in the middle. This pattern holds true across different $L$ values, indicating that the primacy and recency effects are an emergent phenomenon in our WM model. Thirdly, we observed an asymmetry in the strength of the primacy and recency effects, with the primacy effect being more prominent than the recency effect. This is evident when comparing accuracy at positions 1 and 9 for $L = 9$. Furthermore, the primacy and recency effects become more pronounced with longer list lengths $L$, suggesting that these effects are more prominent when the memory load is high. For instance, the effects are minimal for $L = 3$ but become significant for $L = 7$. Fifthly, we compared the behaviors of WM models with humans and found that, though the models were never trained on human data, they approximate human-like qualitative trends in the primacy and recency effects across different serial positions and $L$ values.

Furthermore, we conduct three ablations on VSR task to validate if these findings generalize across modalities, longer list lengths, and training paradigms. First, we report results for LSTM-256 in the Language Serial Recall (LSR) task, which is a language-based variant of the VSR task (**Appendix A3.1**, **Fig. A2B**). Secondly, we experiment with the LSR task containing trials with even longer list lengths (**Appendix A3.2**, **Fig. A2C**). Thirdly, instead of a joint training regime on all the tasks, we conduct individual training of a model only on the VSR task (**Appendix A3.3**, **Fig. A2A**). In all ablations, we observe similar primacy and recency effects suggesting that these findings are general across different modalities, longer list lengths, and various training paradigms. To further explore the underpinning mechanism attributing to intriguing and emergent primacy and recency behaviors in computational models, we expanded our discussions in **Appendix A4.2**.

**B. Short retention intervals have minimal effects on working memory.** We report the accuracy as a function of serial positions under the conditions of different retention intervals in the VIR task (**Sec. 3**, **Fig. 4B**). Aligning with human results, we observe that short retention intervals have minimal effects on WM performances, as indicated by the small performance gap between $T_{retent}$ of 0 and 5 (blue versus orange lines). It is possible that the effect might be more dramatic with longer retention intervals. Moreover, same as **Fig. 4A**, here, we noted that there also exists a small recency effect for both models and humans. In particular, the effect is slightly stronger in $T_{retent} = 5$ than $T_{retent} = 0$. However, this effect is less prominent than the ones observed in **Fig. 4A**, probably due to the short $T_{retent} = 5$ or the small list length.

Moreover, we experimented on retention intervals of 2, 4, and 6 (shown in **Appendix A3.5**). We observed similar results as **Fig. 4B** with no significant performance differences across various short retention intervals (**Fig. A3**).In addition to the duration of retention intervals, we also studied the effect of the stimulus complexity during $P_{retent}$ on the working memory performances. We introduced a variant of the Change Detection (CD) task where we replaced gray images with random noise images (see **Appendix A3.4** for more details).

**C. Domain conflicts and increased cognitive loads impair working memory performances.** To investigate whether domain conflicts would impair WM performance, we report accuracy in the CS task as a function of cognitive loads in different conditions involving combinations of visual and spatial domains (**Sec. 3**, **Fig. 4C**). In humans, regardless of the domain of the distractor task, we observe a monotonic decrease in accuracy with increasing cognitive load. This suggests that WM performance is impaired due to the cognitive load imposed by the distractor tasks, rather than the specific domain of the distractor task itself. However, in the WM model, we observe a decrease in accuracy only when the distractor task is of the spatial domain, implying that our spatial distractor task introduces a higher cognitive load than the visual distractor task, thereby hurting WM performance.

**D. Monotonic decrease in accuracy with increasing set size.** We report the agent's accuracy as a function of set sizes $S$ in SMU task (**Sec. 3**) in **Fig. 4D**. For recurrent models, the recall accuracy monotonically decreases with increasing set sizes. For different recurrent backbones with the same memory capacities, GRU outperforms LSTMs and RNNs (compare GRU-96 vs RNN-96 vs LSTM-96). Moreover, for the same recurrent backbone, larger memory capacity enhances overall recall accuracy across all set sizes (RNN-96 vs RNN-256). Humans show a similar qualitative set size effect as recurrent models. In contrast, TRF-96 shows an opposite trend to humans and recurrent models, with increased accuracy as a function of increasing set size initially. Also, note that TRF-96 outperforms other RNN models in recall accuracy at set size 8 even with smaller memory capacity (e.g. RNN-256).

**E. Working memory stores fine-grained details of visual patterns.** We report the accuracy as a function of serial positions over various pattern-distractor differences $n$ in VSRec (**Sec. 3**, **Fig. 4E**).

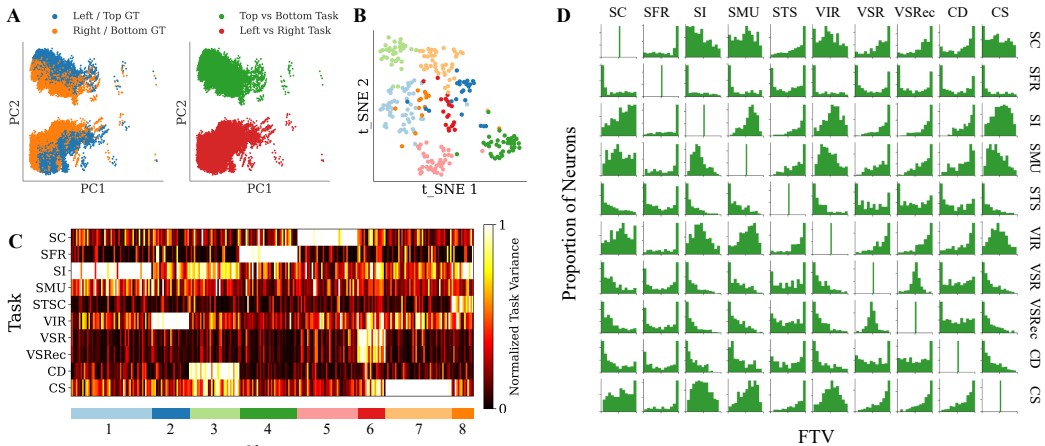

Figure 5: **Visualization of neural correlates and task-specialized neural clusters** (A) For LSTM-1024 in STS task, the visualization results of neural clusters based on response behaviors (left) and task identities (right) are presented. See **Sec. 5G** for details. In (B), we present the t-SNE visualization of neural clusters based on Task Variance (TV) of all hidden units in LSTM-256 over all 10 tasks. In (C), we present the neural selectivity of all hidden units for all the tasks. The neural selectivity is defined as TV values. All hidden units from left to right are sorted based on the clusters in (B). See the colorbar for TV values. In (D), we present the matrix of histogram plots for any pairs of tasks, where each histogram indicates the number of hidden units more selective to one task over the other. The selectivity for each hidden unit over a pair of tasks is defined as the fractional task variance (FTV). The x-axis and the y-axis of the histogram denote the FTV and the frequency of hidden units respectively. See **Sec. 5H** for details.

Intuitively, as it becomes easier to discern the differences between the correct pattern and the distractor, the more accurate it is to recall the correct patterns. Indeed, we see an increase in overall accuracy for bigger $n$. This trend is similar to human results. Moreover, as also observed in **Fig. 4A**, we see a primacy effect on both WM models and humans, where the accuracy is highest at position 1 and slowly decreases over subsequent positions. Interestingly, despite the difficulty of VSRec, for both humans and the models, the accuracy of recalling 36 cells correctly from the distractors with only 2 cell differences is still way above chance. This suggests that both humans and WM models are very good at memorizing fine-grained details of complex patterns.

**F. Memory capacity is independent of the features or conjunctions of features.** We study the capacity for storing different features and conjunctions in the CD task based on the agent's accuracy over set sizes under different feature conditions (**Sec. 3**, **Fig. 4F**). Humans display minimal accuracy differences across feature conditions, suggesting that memory capacity is independent of specific features or feature combinations. Similarly, WM models exhibit comparable accuracy regardless of feature conditions. However, caution is advised in interpreting model results, as accuracy saturates at 100% regardless of set size, likely due to overfitting. To draw stronger conclusions, further investigation in a low-data regime using our trials is needed.

**G. Neural population carries active task representation in task-switching experiment.** In STS task (**Sec. 3**), we present the visualization results of clusters based on neural activation of all hidden units across all trials and all conditions for LSTM-1024 in **Fig. 5A**. Specifically, we first take the hidden representation during $P_{probe}$ for all the trials within the task and then perform t-SNE [58] to project the hidden representation to 2D. On the left panel in **Fig. 5A**, we color-code the points based on the ground truth at that particular $t_{probe}$. In this case, we did not observe distinct colored clusters whereas when we color-coded the points based on the task that was supposed to be performed at $t_{probe}$, we observe two distinct colored clusters. In other words, the neural representations in the recurrent model encode task identities within a task-switching paradigm, thus playing a supervisory role. See **Appendix A4.3** for more details.

**H. Responses from neural populations reveal task selectivity during joint training.** To study the task-specialized neural clusters in recurrent models, we present the visualization results in **Fig. 5B-D**. See **Appendix A4.4** for the steps to obtain and interpret the visualization results. From **Fig. 5B, C**, we observed 8 distinct neural clusters and each neural cluster is selective to a different set of tasks.

For example, cluster 4 specializes in the SFR task, while cluster 6 is active in temporal domain tasks like VSR and VSRec. We also note that clusters of neurons mostly emerge in networks of lower capacity with memory constraints. From **Fig. 5D**, we show diverse neural populations where some have preferences for one task over the other, while some are equally active in both tasks. For instance, the entry in row 1, and column 5 indicates that the majority of hidden units are more selective to SC task compared with STS, as the distribution of neural population shifts to the right with the majority of the neurons having FTV = 1. Additionally, we conduct experiments where we lesion specific neural clusters to investigate causal evidence. As expected, we observe the corresponding performance drops in the accuracy of specific tasks depending on the neural cluster that has been lesioned (shown in **Fig. A6**). See **Appendix A4.5** for details.

# 6 Discussion

Despite significant research progress in studying individual aspects of working memory (WM), there remains a huge gap in the broader study of WM. We take initial steps in this direction by establishing a systematic and quantitative methodology to study the multi-facets of WM. However, it is important to recognize that we are still at the preliminary stage in a broader exploration of WM. The complexity, flexibility, and variability of WM present numerous avenues for future research. These avenues may involve refining the temporal alignment between human performance and model predictions, fine-tuning task difficulty across experiments by introducing variations in experimental parameters, experimenting with different training and testing methods, investigating the models' generalization abilities across a broader spectrum of tasks and modalities, and extending benchmarking experiments to include bio-inspired working memory models.

In our work, we introduce a comprehensive benchmark dataset consisting of 10 tasks, and 1 million trials covering 4 functionalities and 3 domains of WM. Moreover, we benchmark WM models and humans on the 10 tasks and compare their 11 memory characteristics at the behavioral, performance, and neural levels with a set of newly introduced evaluation metrics. We obtained insights about the underlying mechanism of WM from these tasks and identified the limitations in existing WM models.

It is worth noting that all the models were never trained with any human data or behavioral biases. Yet, these models still remarkably exhibit interesting alignments with human behaviors, such as in dependency on set sizes, pattern similarities, and primacy and recency effects. We report the top-1 accuracy for each condition in all the experiments; however, caution is needed when comparing the absolute performances between WM models and humans for multiple reasons: (1) differences in the concept of time, with humans perceiving time continuously in seconds while AI operates in discrete time steps; (2) varied learning mechanisms, including Hebbian learning in biological systems and back-propagation in AI; (3) distinct neuronal mechanisms between artificial and biological neurons, such as synaptic plasticity and structural plasticity; (4) disparities in the visual diets fed to humans and AI models, with humans exposed to video streams and AI models working with random static images; and (5) constraints on memory capacity varying between humans and AI models.

Our work contributes a suite of tools with detailed documentation for stimulus synthesis and parameter configurations in psychophysics experiments, training and testing state-of-the-art working memory models, evaluating model behaviors and neural responses, and comparing behavioral consistency between humans and models. This methodology serves as a valuable resource for comprehensive investigations into working memory, fostering advancements in both biological and artificial intelligence systems.

## Acknowledgments and Disclosure of Funding

This research is supported by the National Research Foundation, Singapore under its AI Singapore Programme (AISG Award No: AISG2-RP-2021-025), its NRFF award NRF-NRFF15-2023-0001, Mengmi Zhang's Startup Grant from Agency for Science, Technology, and Research (A*STAR), and Early Career Investigatorship from Center for Frontier AI Research (CFAR), A*STAR. The authors declare that they have no competing interests.

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

## A1  Detailed Descriptions about Psychophysics Experiments in WorM

**Experiment A1 - Spatial Free Recall (SFR) [13]**. This experiment focuses on the spatial domain of WM and aims to assess the ability to remember spatial locations and engage in immediate free recall of that information. **Experiment schematic:** The experiment begins with $P_{present}$ followed by $P_{probe}$. In each trial, the stimulus image in $P_{present}$ contains 30 green squares randomly chosen from a $10 \times 10$ grid. At every $t_{present}$, one randomly selected square of the 30 green ones turns red. Finally, in $P_{probe}$, a grey blank image is presented signaling the WM model to recall all the past spatial locations where the squares turned red. On the other hand, humans received an auditory cue to begin the recall. Humans indicate the highlighted squares by clicking on them in any order using a mouse cursor. The WM models, however, engage in a 100-way multi-label classification task with multi-hot encoded targets corresponding to the highlighted square locations. Afterward, we select the top-$L$ squares from the WM model's prediction based on the output probabilities, sort them in descending order, and generate the model's recall response, where $L = T_{present}$ refers to the list length in the trial. **Experiment conditions:** We vary list length $L = T_{present} = 1 - 8, 10, 12, 15, 18$. Refer **Fig. A1A** for example trial.

**Experiment A2 - Change Detection (CD) [35]**. This experiment involves the visual domain of WM and aims to evaluate the agent's memory capacity for simple features and conjunctions. **Experiment schematic:** The experiment consists of $P_{present}$, $P_{retent}$, and $P_{probe}$ in the respective order. In $P_{present}$ where $T_{present} = 1$, a memory array, consisting of $S$ bars with varying attributes in color (red or green), orientation (horizontal or vertical), size (small or big), and presence/absence of a gap in the middle of each bar, is presented. Here, $S$ refers to the set size of the trial. During $P_{retent}$, a series of blank grey images are presented. In $P_{probe}$, where $T_{probe} = 1$, agents are probed with a test array that can be either identical to the memory array or has one bar that differs along a single feature dimension. The agents have to make a binary prediction about whether or not there is a change given the test array. **Experiment conditions:** We formulate 5 main experiment conditions by varying the change of the test array in five aspects: color, orientation, bar size, presence or absence of the gap, and conjunctions of the four features where either of the four features could change. Additionally, we vary S from 2, 4, 6, 8, 10, 12 and also vary $T_{retent}$ from 0, 6, 12, and 18. See **Fig. A1B** for an example trial from this experiment.

**Experiment A3 - Visual Item Recognition (VIR) [29]**. Here, we examine visual WM with different length lists and investigate the effect of retention interval on recognition performance. **Experiment schematic:** The experiment is structured with three phases presented in the following order: $P_{present}$, $P_{retent}$, and $P_{probe}$. In $P_{present}$, a series of distinct matrix patterns are presented in sequence. These patterns are composed of $6 \times 6$ square cells, with half of the cells being filled with red. In $P_{retent}$, a series of grey blank images are presented. In $P_{probe}$, where $T_{probe} = 1$, a probe image is presented, consisting of two patterns presented side-by-side. One of these patterns is an exact match to one of the previously shown patterns, while the other pattern serves as a distractor. The distractor pattern is generated from the target pattern by changing 2 unfilled cells to filled ones and vice-versa. The agents have to perform a binary classification task, wherein 0 represents the left pattern and 1 represents the right pattern. **Experiment conditions:** We vary list length $L = T_{present}$ from 4, 6, 8, and 10, and also vary $T_{retent}$ from 0, 2, 4, 5, and 6. See **Fig. A1C** for an example trial.

**Experiment A4 - Visual Serial Recall (VSR) [4]**. This experiment evaluates the ability to accurately recall matrix patterns as well as the order in which they were presented. Therefore, this experiment encompasses both the visual and temporal domains of WM. **Experiment schematic:** The experiment consists of $P_{present}$ followed by $P_{probe}$. $P_{present}$ involves the presentation of $L$ matrix patterns in a sequential manner. These matrix patterns consist of square cells arranged in a $6 \times 6$ grid, with half of the cells filled in green. In $P_{probe}$, a probe image is displayed containing all the $L$ patterns presented during $P_{present}$. These $L$ patterns are placed simultaneously at randomly chosen locations from a set of 9 evenly spaced locations in a $3 \times 3$ grid. Locations that were not used remained blank. At each $t_{probe}$, the agents perform a 9-way classification task to indicate the first pattern presented during $P_{present}$, followed by the second pattern, and so on. **Experiment conditions:** We vary list length $L = T_{present}$ from $2 - 9$. An example trial is shown in **Fig. A1D**.

**Experiment A5 - Visual Serial Recognition (VSRec) [4]**. In this experiment, agents are required to recall sequentially presented matrix patterns in a forward serial order within a recognition paradigm. Here, we also investigate the resolution of the recalled patterns by varying the difference between target patterns and distractor patterns. This experiment again involves both the visual and temporal

domains of WM. **Experiment schematic:** The experiment is similar to the VSR experiment above in terms of the setup. The experiment consists of $P_{present}$ and $P_{probe}$. $P_{present}$ is exactly the same as $P_{present}$ in the VSR experiment. The only differences happen at $P_{probe}$. During $P_{probe}$, a series of two-alternative recognition tests are presented in consecutive time steps to assess memory for the presented patterns in forward serial order. Essentially, at every $t_{probe}$, the target pattern from corresponding $t_{present}$ and a distractor pattern are presented side by side. The agents perform a binary classification task to indicate the target pattern, where 0 corresponded to the left pattern and 1 corresponded to the right pattern. The distractor patterns were generated from the target pattern by changing the values of n cells such that the total number of filled cells remained constant. We denote $n$ as the pattern-distractor difference. We made sure that the distractor patterns were not among the initially presented matrix patterns. **Experiment conditions:** We vary list length $L = T_{present}$ from 2 to 9 and the pattern-distractor difference $n$ from the range of 2, 4, 6, 8, and 10. See **Fig. A1E** for an example trial.

**Experiment A6 - Complex Span (CS) [60]**. This experiment aims to investigate the interference between the visual and spatial domains of WM, as well as the effect of cognitive load on recall performance. Moreover, in this experiment, agents are required to perform a serial recall of information. Hence, this experiment encompasses all three domains of WM: visual, spatial, and temporal. **Experiment schematics:** The experiment consists of two types of $P_{present}$ and two corresponding types of $P_{probe}$, and two types of $P_{distractor}$. (I) The first type of $P_{present}$ and $P_{probe}$ corresponds to visual storage and involves memorizing a sequence of visual patterns in $P_{present}$ and recalling the exact visual patterns in a serial order in $P_{probe}$. The visual patterns are represented as 4 $\times$ 4 grid, where half of the cells are in red. (II) The second type of $P_{present}$ and $P_{probe}$ corresponds to spatial storage and involves memorizing a sequence of spatial patterns involving ball movements in $P_{present}$ and recalling the exact directions of ball movements in a serial order in $P_{probe}$. To interfere with the spatial/visual memories in $P_{present}$, two types of $P_{distractor}$ are introduced: (III) The first type of $P_{distractor}$ corresponds to the visual domain and involves color discrimination tasks where agents have to classify the color of the given panel as blue or red. (IV) The second type of $P_{distractor}$ corresponds to the spatial domain and involves symmetry discrimination tasks where agents have to classify whether the given pattern is symmetric about the vertical axis or not. **Experiment conditions:** We consider four different variations: I($P_{present}$) + III($P_{distractor}$) + I($P_{probe}$); II($P_{present}$) + III($P_{distractor}$) + II($P_{probe}$); I($P_{present}$) + IV($P_{distractor}$) + I($P_{probe}$); and II($P_{present}$) + IV($P_{distractor}$) + II($P_{probe}$); namely, visual + visual, spatial+visual, visual+spatial, and spatial + spatial respectively. Additionally, we change the cognitive load by varying $T_{distractor}$ from 0, 1, 3, and 5. See **Fig. A1F** for an example trial.

**Experiment B1 - Spatial Coordination (SC) [41]**. The objective of this experiment is to assess the agents' working memory (WM) capacity in the spatial domain, specifically focusing on their ability to coordinate and integrate spatial information across various time steps. **Experiment schematics:** The experiment consists of $P_{present}$ and $P_{probe}$ in the respective order. At every $t_{present}$, one cell in a 10 $\times$ 10 grid is highlighted in green. In $P_{probe}$, the agents are probed using a blank grey image to perform a binary discrimination task to indicate whether the pattern integrated mentally over all $T_{present}$ is symmetric about the vertical axis or not. **Experimental conditions:** We vary the list lengths $L = T_{present}$ from 10, 12, 14, 16, and 18. Refer **Fig. A1G** for an example trial.

**Experiment B2 - Spatial Integration (SI) [52]**. This experiment focuses on the spatial domain and aims to investigate the agents' abilities to integrate spatial information. **Experiment schematics:** The experiment consists of $P_{present}$ and $P_{probe}$. In $P_{present}$, there is a sequential presentation of partial line drawings at every $t_{present}$. In $P_{probe}$, where $T_{probe} = 1$, a complete line drawing containing 12 line segments is presented. The task is to mentally integrate the partial drawings shown during $P_{present}$ and compare them to the complete line drawing shown during $P_{probe}$. Essentially, the agents perform a binary classification to indicate whether the figure shown in $P_{probe}$ is identical to the mentally integrated figure from $P_{present}$. All line drawings were generated by connecting neighboring points within an imaginary 4 $\times$ 4 grid. **Experiment conditions:** We vary the number of line segments in the partial drawings from 12, 6, 4, 3, 2, and 1, resulting in list lengths $L = T_{present}$ of 1, 2, 3, 4, 6, and 12 respectively. As the list length increases, the number of integrations needed to solve the trial also increases. See **Fig. A1H** for an example trial.

**Experiment C1 - Spatial Memory Updating (SMU) [44]**. This experiment assesses the capabilities of memory manipulation in the spatial domain. **Experiment schematics:** The experiment consists of $P_{present}$, $P_{update}$, and $P_{probe}$ in the respective order. In $P_{present}$ where $T_{present} = 1$, a stimulus is

presented, consisting of $S$ green squares, arranged in an imaginary circular formation. Within each square, there is a red marker located in one of the nine possible locations within an invisible $3\times3$ grid. Agents are required to memorize the spatial locations of the markers within each square. Next, at every $t_{update}$ in $P_{update}$, an arrow is presented at the center of one of the squares, indicating the direction for mentally updating the marker's new location. The arrows can be oriented vertically, horizontally, or diagonally, with the condition that the marker never exits the square. A sequence of a total of 8 memory update operations is presented in a clockwise sequence through the $S$ squares. Hence, $T_{update} = 8$. Finally, at every $t_{probe}$ of $P_{probe}$, one out of the $S$ squares is highlighted in red. This red square acts as a cue for the agents to recall and indicate the marker's location within that particular square through a 9-way classification task. The red marker locations for all $S$ squares are probed in random order until $t_{probe} = S$. **Experiment conditions:** We vary $S$ from 1 to 8. Refer **Fig. A1I** for an example trial.

**Experiment D1 - Spatial Task Switching (STS) [38]**. This experiment involves the spatial domain of WM and investigates the agent's capacity to flexibly transition between two spatial discrimination tasks based on supervision cues. **Experiment schematics:** The experiment consists of two types of phases $P_{switch}$ and $P_{probe}$ presented in random order. In $P_{switch}$, a cue image is presented to the agents to indicate the discrimination task to be performed for subsequent $P_{probe}$. There are two binary discrimination tasks: top-versus-bottom and left-versus-right. For the left-versus-right discrimination task, the cue image contains dash markers placed on the left and right sides of the image. In contrast, for the top-versus-bottom discrimination task, the dash markers are positioned on the top and bottom sides of the image. At every $t_{probe}$ in $P_{probe}$, a red marker is presented in any of the four locations within a $2 \times 2$ grid. The agents are required to switch between the two discrimination tasks during every $P_{switch}$ and respond with 0 for top/left positions and 1 for bottom/right positions. **Experiment conditions:** We randomly vary the number of task switches i.e. $P_{switch}$ in different trials as well as the time step at which task switches occur. See **Fig. A1J** for an example trial.

**Experimental trial splits**. In each WM psychophysics experiment, we generate a total of 86,400 trials for training, 9,600 trials for validation, and 9,600 trials for testing. Importantly, note that for the CD experiment, there are five distinct conditions, each of which consists of its own set of 86,400 trials for training, 9,600 trials for validation, and 9,600 trials for testing.

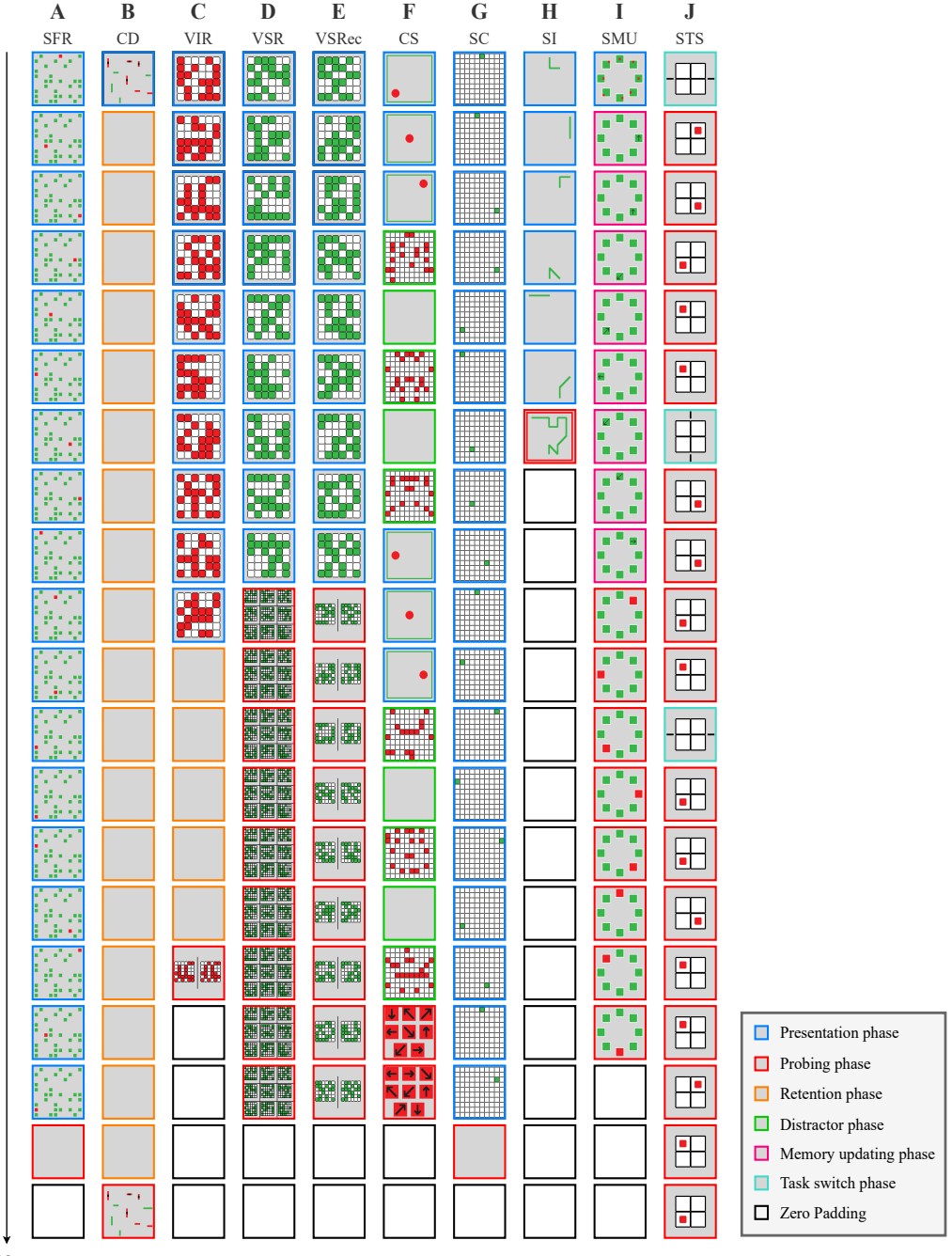

Figure A1: **Detailed schematic of each experiment.** We expand the overviews of all ten experiments introduced in **Fig. 1** and **Sec. 3** with detailed experiment schematics. Each column represents the schematic of one experiment. The time goes from the top to the bottom. Each row presents the example stimulus at that time step $t$. The legend and interpretations of this figure follow **Fig. 1**.

## A2 Computational Models of Working Memory

### A2.1 Additional Implementation and Training Details

**Feature Extractor**. In all our models, we employ a 4-layer 2D-convolutional network with 64, 128, 256, and 512 channels in the successive layers. We also experimented with ViT [21] pre-trained on ImageNet, where we freeze the model and use it for feature extraction. However, features from natural images of ImageNet [18] do not generalize to synthetic images of our dataset.

**Task Embeddings**. We learn task-specific embeddings $M \in \mathbb{R}^{14 \times 14}$, which informs the model of which task needs to be performed. There are a total of 10 WM tasks. Although, the Change Detection (CD) task comprises five distinct task conditions, namely color, orientation, size, gap, and conjunction, each of which necessitates a unique task identifier. Consequently, there are a total of 14 (9+5) unique task embeddings.

**Training Details**. During training, we sample 10 trials from each task, obtain the model's responses for all the sampled trials, and compute task-specific losses for each respective task. These individual losses are then aggregated and the joint loss is utilized to perform back-propagation. In essence, the model parameters are optimized jointly over all tasks. Note that we formulate the SFR task as multi-label classification and the other tasks as multi-class or binary classification. Consequently, the task-specific losses are computed using either cross-entropy or binary cross-entropy, depending on the nature of the tasks.

**Implementation Details**. All our models were trained from scratch. Hyperparamater tuning was done using grid search. We show the different hyperparameters for our models in **Tab. A1**. We employed the Adam optimizer [30] with a starting learning rate of 1e-4 and the first moment of 0.9. The learning scheduler reduces the learning rate by a factor of 0.8 when the validation loss doesn't improve for 3 epochs. We report the average top-1 accuracy and standard error for various model architectures and capacities, computed across test trials from different conditions. For all our experiments, we used 16 NVIDIA RTX A5000 GPUs, each equipped with 24 GB of memory.

**Data Generation Pipeline.** Given any condition in a task, we follow the three steps below to synthesize stimulus for every time step. For an easy illustration of the three steps, we use the Visual Serial Recognition task (**Appendix A1**) as an example. First, depending on the list length of the trial, we generate multiple unique $6 \times 6$ matrix patterns by sampling grid cells to be filled. These matrix patterns constitute the memory and distractor items for the trial. Second, we generate the ground truth labels for $\text{P}_{probe}$ based on the sampled matrix patterns. Third, given all the patterns sampled in Step 1, we synthesize the stimulus using the Pillow library [11].

Our systematic process ensures the creation of task-specific trials with multiple controlled variations and difficulty. We provide detailed documentation and made our code easy to use and customize for future studies, such as generalization tests of working memory models with longer task sequences, and larger set sizes. Our code and data are publicly available: link.

### A2.2 Time mapping between WM models and human psychophysics experiments

There could be multiple ways of establishing the time mapping between WM models and human psychophysics experiments (**Fig. 1**). Here, for the WM models, we stick to 1 stimulus per time step rule. In **Tab. A2**, we report the mapping of a single time step in WM models to the duration of the corresponding stimulus presentation in human experiments. For instance, in the SFR human

| Hyperparameters | RNN | GRU | LSTM | TRF |
|---|---|---|---|---|
| Input Size | 512 | 512 | 512 | - |
| Hidden / Embedding Size | [96, 256, 1024] | [96, 256, 1024] | [96, 256, 1024] | [96, 256, 1024] |
| TRF Encoder Layers | - | - | - | 2 |
| TRF Attention Heads | - | - | - | 8 |
| Batch Size | [**140**, 70] | [**140**, 70] | [**140**, 70] | [**140**, 70] |
| Learning Rate | [0.00008, **0.0001**, 0.0003] | [0.00008, **0.0001**, 0.0003] | [0.00008, **0.0001**, 0.0003] | [0.00008, **0.0001**, 0.0003] |
| Epochs | 200 | 200 | 200 | 200 |

Table A1: **Different hyperparameters considered for models trained on the WorM dataset.** The best hyperparameters for each model are in bold. Here, batch size refers to the total number of trials from all tasks within one batch.

| TASK | $P_{present}$ | $P_{retent}$ | $P_{probe}$ | $P_{distractor}$ | $P_{switch}$ | $P_{update}$ |
|------|---------------|--------------|-------------|------------------|--------------|--------------|
| SFR | 750 ms | * | - | * | * | * |
| CD | 100 ms | 150 ms | 2000 ms | * | * | * |
| VIR | 1000 ms | 1000 ms | - | * | * | * |
| VSR | 1550 ms | * | - | * | * | * |
| VSRec | 1550 ms | * | - | * | * | * |
| CS | 500 ms | * | - | 1700 ms | * | * |
| SC | 1000 ms | * | - | * | * | * |
| SI | - | * | - | * | * | * |
| SMU | - | * | - | * | * | - |
| STS | * | * | 1500 ms | * | 203 ms | * |

Table A2: **Time mapping between WM models and Human psychophysics experiments.** We show the mapping of a single time step in WM models to **ms** in different phases of the corresponding human psychophysics experiments. Asterisks (*) indicate the absence of a specific phase in the experiment. Dashes (-) indicate that either humans were provided with an indefinite amount of time during that phase, or the duration of stimulus presentation was variable and not standardized.

experiment, the duration of each stimulus presentation during $P_{present}$ was 750 ms, whereas the WM models were exposed to each stimulus for only 1 time step. Therefore, within the $P_{present}$ phase of the SFR experiment, a single time step of the WM models corresponds to a duration of 750 ms in the human experiments.

### A2.3 Quantitative comparison with human behaviors

To quantitatively compare computational models and humans, we introduce two types of scores: the slope difference score ($A$) and the average accuracy difference score ($B$). First, we define the slope difference score ($A$), as $|(M_A - H_A)/H_A|$, where $M_A$ and $H_A$ are slopes for computational models and humans respectively. We use the slope difference score to capture similarity in the trend of accuracy against different conditions like serial position, set size, and cognitive load. Second, we define the average accuracy difference score ($B$), as $|(M_B - H_B)/H_B|$, where $M_B$ and $H_B$ are average accuracy for computational models and humans respectively across different conditions. For example, $A_{patterndiff}$ in VSRec task (**Fig. 4E**) for LSTM-256 model is calculated by first computing $M_A$ i.e. the slope of the best-fit linear line for the model's top 1 accuracy as a function of serial positions in the plot. Likewise, we compute $H_A$ and hence, $A_{patterndiff}$. Similarly, to calculate $M_B$, we first compute the average accuracy of the model over all the serial positions for each condition of pattern difference and then compute the slope of the best-fit linear line for the model's top 1 accuracy as a function of pattern differences in the plot. Similar calculations follow afterward to obtain $H_B$ and subsequently $B_{patterndiff}$. Note that in the VSR task (**Fig. 4A**), we calculate two different scores for the primacy ($A_{primacy}$) and recency ($A_{recency}$) effect. For this, we divide the list at the point of minimum top-1 accuracy, into two lists, i.e. primacy and recency lists.

With these evaluation metrics, we report the quantitative measure of behavioral consistency with humans in **Tab. A3**. Ideally, if the model performs the same as humans, the slope difference would be zero. To test whether the results are significantly different from the chance model, we computed the p-value using two-tailed t-tests. From the results, we found that the majority of the models are more consistent with humans than the chance model; however, there still exists performance gaps between these models and humans.

| | R-96 | R-256 | R-1024 | G-96 | G-256 | G-1024 | L-96 | L-256 | L-1024 | T-96 | T-256 | T-1024 |
|---|---|---|---|---|---|---|---|---|---|---|---|---|
| **VSR Task** | | | | | | | | | | | | |
| $A_{primacy}$ | 3.300* | 3.264* | 3.112* | 8.108* | 6.748* | 6.859* | 3.363* | 6.458* | 6.538* | 3.307* | 3.287* | 3.480* |
| $A_{recency}$ | 1.164* | 1.183* | 1.138* | 1.449* | 1.327* | 1.276* | 1.134* | 1.384* | 1.328* | 1.170* | 1.124* | 1.102* |
| $B_{listlength}$ | 0.692* | 0.697* | 0.705* | 0.467* | 0.387* | 0.400* | 0.696* | 0.348* | 0.364* | 0.693* | 0.693* | 0.693* |
| **VIR Task** | | | | | | | | | | | | |
| $A_{retentioninterval}$ | 0.324* | 0.729* | 0.958* | 0.801* | 2.244* | 0.729* | 3.021* | 0.951* | 0.554* | 0.995* | 0.968* | 1.438* |
| $B_{retentioninterval}$ | 0.245* | 0.272* | 0.261* | 0.164* | 0.126* | 0.068* | 0.265* | 0.057* | 0.140* | 0.446* | 0.441* | 0.262* |
| **CS Task** | | | | | | | | | | | | |
| $A_{cognitiveload}$ | 0.995* | 0.997* | 0.996* | 0.995* | 0.996* | 0.989* | 0.999* | 0.998* | 0.991* | 0.998* | 0.999* | 0.995* |
| $B_{cognitiveload}$ | 0.754* | 0.763* | 0.718* | 0.750* | 0.741* | 0.350* | 0.708* | 0.594* | 0.340* | 0.218* | 0.511* | 0.773* |
| **SMU Task** | | | | | | | | | | | | |
| $A_{setsize}$ | 0.347* | 1.059* | 0.006* | 1.001* | 1.001* | 1.000* | 0.985* | 0.993* | 0.999* | 1.049* | 1.000* | 1.255* |
| $B_{setsize}$ | 0.680* | 0.396* | 0.310* | 0.160* | 0.160* | 0.161* | 0.159* | 0.160* | 0.161* | 0.155* | 0.161* | 0.133* |
| **VSRec Task** | | | | | | | | | | | | |
| $A_{patterndiff}$ | 1.103* | 1.382* | 0.979* | 0.150* | 0.569* | 0.910* | 0.531* | 0.097* | 0.625* | 0.977* | 0.923* | 1.196* |
| $B_{patterndiff}$ | 0.302* | 0.318* | 0.347* | 0.030* | 0.271* | 0.311* | 0.080* | 0.220* | 0.287* | 0.356* | 0.358* | 0.323* |
| **CD Task** | | | | | | | | | | | | |
| $A_{setsize}$ | 0.999* | 0.999* | 0.799* | 1.003* | 0.999* | 0.999* | 0.999* | 0.999* | 0.999* | 0.999* | 0.999* | 1.003* |
| $B_{setsize}$ | 0.110* | 0.110* | 0.154* | 0.110* | 0.110* | 0.110* | 0.110* | 0.110* | 0.110* | 0.110* | 0.110* | 0.110* |

Table A3: **Quantitative scores for comparing humans and computational models on different behavioral effects.** We conduct two-tailed t-tests against chance and compute corresponding p-values. Asterisk (*) indicates p-value < 0.01. See **Sec. A2.3** for introduction to slope difference score $A$ and average accuracy difference score $B$.

## A3 Ablation Experiments

### A3.1 Experiments for language modality

We introduce a new experiment in language modality, with a paradigm similar to that of Visual Serial Recall (VSR) **Sec. 3**. Here, we investigate the behavioral characteristics of the computational models in language modality instead of the visual modality.

**Language Serial Recall (LSR)**. This experiment assesses the ability to recall words accurately as well as the order in which they were presented. **Experiment schematic:** The experiment consists of $P_{present}$ followed by $P_{probe}$. $P_{present}$ involves the presentation of $L$ words in a sequential manner. We randomly sample these words from the Google News dataset. This dataset contains pre-trained vectors for around 3 million words and phrases. At each time step during $P_{present}$, we first map the word to the corresponding 300-dimensional word embedding followed by zero padding this vector to produce $L \times 300$ dimensional representation which is then fed to the WM model. In $P_{probe}$, at each time step, all the $L$ words are presented simultaneously. We collect the embeddings for all $L$ words and append them in a random order to generate $L \times 300$ dimensional representation. At each $t_{probe}$, based on this representation, the agents perform a 9-way classification task to indicate the first word presented during $P_{present}$, followed by the second word, and so on. **Experiment conditions:** We vary list length $L = T_{present}$ from $2 - 9$.

**Result analysis:** From the experimental results shown in **Fig. A2B**, we observed consistent performances as the ones from the VSR experiment. First, as list length increases, there is a corresponding increase in the memory load. Consequently, we observe a decrease in overall accuracy across various list lengths. We also observe pronounced primacy effects across varying list lengths, alongside a recency effect that becomes apparent in longer list lengths. These results suggest that the behavioral findings generalize across different modalities.

### A3.2 Training with Longer List Lengths

To verify if our behavioral findings are valid for longer list lengths, we train our model on the LSR Task (**Appendix A3.1**) with trials of list lengths up to 16 instead of 9. Therefore, in this case, trials consist of 32 time steps in total instead of 20 time steps. We report our results in **Fig. A2C**. Here, we observe similar primacy and recency effects as we observed in the LSR task with shorter list lengths (compare with **Fig. A2A**). We notice similar trends where primacy effects are prominent in shorter list lengths and the recency effects start emerging in longer list lengths with higher memory load. This result suggests that the primacy-recency effect is a general phenomenon even for longer list lengths.

### A3.3 Individual task training

To isolate the effect of the joint training paradigm, we also conduct individual training on the VSR task. Specifically, we train the LSTM-256 model only on the VSR task. We report the results in **Fig. A2A**. We make similar observations as in the joint training paradigm. First, with increasing list length, the overall accuracy of the model decreases. Next, the primacy-recency effect is more prominent in cases of longer list lengths with higher memory load. Finally, the primacy effect is more prominent in smaller list lengths while the recency effect is observed for longer list lengths. This indicates that the primacy-recency effect is a more general phenomenon agnostic of the training paradigm.

### A3.4 Increased difficulty in CD task

In order to introduce additional difficulty in the CD task, we also experimented with more complex variants where we presented random noise images during the retention phase, instead of simple gray images. These random noise images are generated by randomly sampling a pixel value from 0-255 for every pixel of RGB channels of the image. This also aligns with the backward masking technique in neuroscience, which is often used for disrupting recurrent connections in the biological brains [49]. We train and test the LSTM-96 model on this new variant of the CD task. We observe that while LSTM-96 was successful in completely solving the initial version of the CD task with 100% accuracy, it completely failed to learn this modified variant.

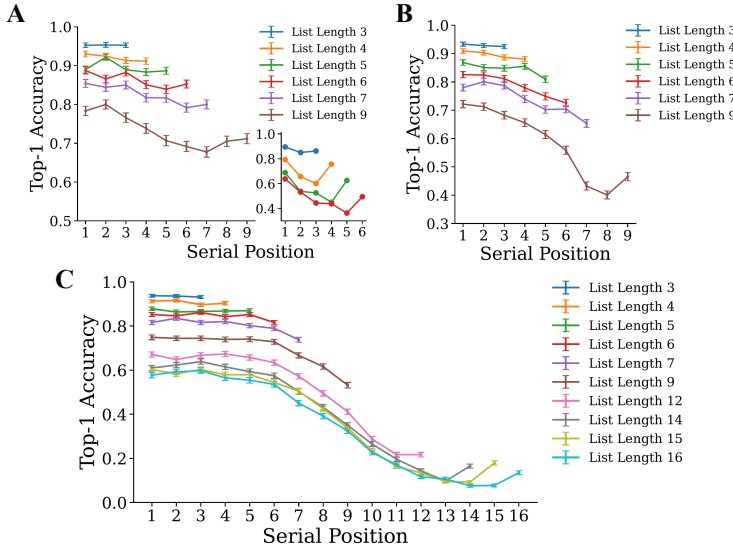

Figure A2: **Ablation experiments.** (A) We show the top-1 accuracy of the LSTM-256 model as a function of the serial position of the memory items in the VSR task. Here, the model was trained only on the VSR task. In (B), we show the behavioral accuracy as a function of list lengths in LSR task for LSTM-256 model. In (C), we show the plot for an LSTM-256 model trained only on the LSR task with longer sequence lengths (32 time steps).

### A3.5 Fine-grained retention interval testing in VIR task

We evaluate the LSTM-1024 model on more retention intervals in the VIR task. Specifically, we test our model on 0, 2, 4, 5, and 6 retention intervals. We show these results in **Fig. A3**. We observe similar results as **Fig. 4B**, and we do not find any statistical significance in working memory performance differences across various short retention intervals.

## A4 Additional Results and Analysis

### A4.1 Behavioral Performance Comparison across Architectures and Memory Capacities

We analyzed various memory characteristics in **Fig. 4**. Here, we expand over different model architectures and model capacities in **Fig A7, A8, A9, A10, A11, A12**. We found that the performances in top-1 accuracy vary across architectures and memory capacities. In general, the larger the memory capacity, the better the behavioral performance. However, there are a few exceptions where the models with smaller capacities yield higher accuracy (e.g. **Fig. A11**, J versus K). We also notice that some tasks are extremely difficult for all WM models to learn regardless of the

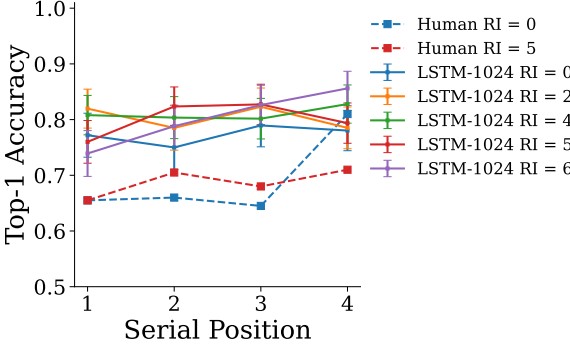

Figure A3: **Short retention interval testing in VIR task.** We show the results for LSTM-1024 on the VIR task with multiple retention intervals including 0, 2, 4, 5, and 6. The chance level is 0.5 for all the above plots.

| Models | Hidden size | Number of parameters |
|---|---|---|
| RNN-96 | 96 | 1.8M |
| RNN-256 | 256 | 2M |
| RNN-1024 | 1024 | 3.5M |
| GRU-96 | 96 | 2M |
| GRU-256 | 256 | 2.5M |
| GRU-1024 | 1024 | 6.5M |
| LSTM-96 | 96 | 2M |
| LSTM-256 | 256 | 2.5M |
| LSTM-1024 | 1024 | 8M |
| TRF-96 | 96 | 2.5M |
| TRF-256 | 256 | 4.5M |
| TRF-1024 | 1024 | 20M |

Table A4: **Different computational models of varying memory capacity with their number of parameters.**

architectures. For example, in **Fig. A9**, almost all the WM models fail. Even with a large memory capacity of 1024, which proves to be sufficient for performing other WM tasks, it still performs badly in the CS task.

For a fair comparison, we can either compare different architectures with an equal number of hidden sizes or with an equal number of network parameters. To compare different architectures with the same hidden size, we compare RNN-1024, GRU-1024, LSTM-1024, and TRF-1024. These models are jointly trained on all the working memory tasks. We show the joint average accuracy, i.e., the average accuracy of these models over all tasks in **Fig. A4**. We observe that LSTM-1024 achieves the best joint average accuracy, followed by GRU-1024, TRF-1024, and RNN-1024 respectively.

Similarly, to compare architectures with the same number of parameters, we can compare RNN-256, GRU-256, LSTM-256, and TRF-96 (see **Tab. A4** for the number of parameters in different models). In **Fig. A4**, we again observe LSTM-256 is the best performing model, followed by GRU-256, TRF-96, and RNN-256 respectively. In either comparison, we can note that recurrent networks like LSTMs and GRUs perform better than transformers. While recurrent architectures (GRUs, LSTMs) generally perform better in top-1 accuracy and exhibit the primacy and recency effect (**Sec. 5A**), the transformer architectures fail to replicate such phenomenon; and their top-1 accuracy is also lower than the recurrent architectures.

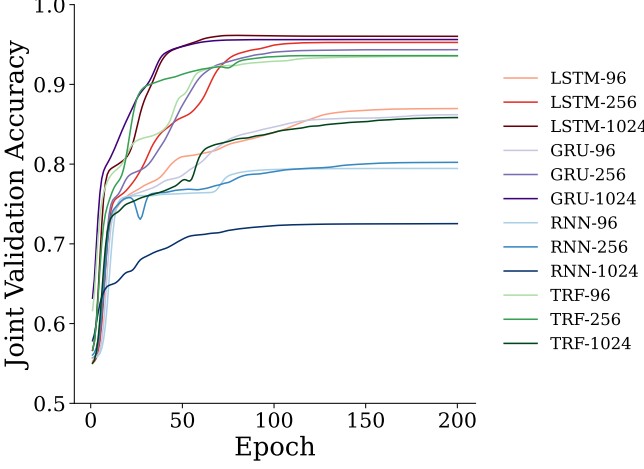

Figure A4: **Training curves for different model architectures and capacities.** The y-axis shows the averaged validation accuracy across all tasks and the x-axis shows epochs.

## A4.2 Further probing into the mechanism attributing to primacy effects

In **Fig. 4A** and **Fig. A2**, we observe consistent primacy effects. Our hypothesis for the primacy effect was that the first stimulus stored in the working memory acts as the first principal component, steering the memory representations of the subsequent stimulus to align with the representation of the first stimulus. Hence, the first item was forgotten less; thus, the network exhibits the primacy effect. To validate this hypothesis, in every trial of the Visual Serial Recall (VSR) task, we first compute the hidden states at time step 1, when the first item is presented, and then compute its cosine similarity with the hidden state representations over subsequent time steps $t$ when subsequent memory items are presented. We report the results in **Fig. A5**. Based on our hypothesis, the cosine similarity between the $1^{st}$ time step with $n^{th}$ time step should be higher than the cosine similarity between $2^{nd}$ time step and $n^{th}$ time step. However, that is not the case in **Fig. A5** (see along column), suggesting that our hypothesis is incorrect. Future work should explore other plausible hypotheses for the primacy effect in recurrent models.

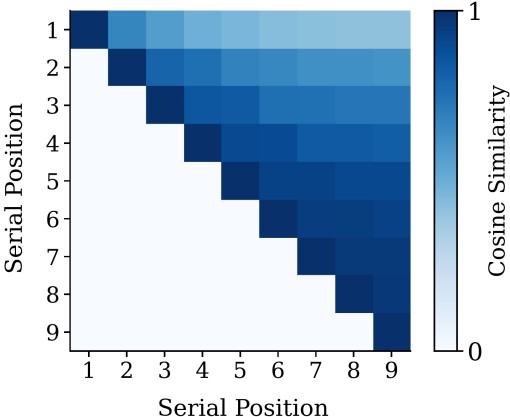

Figure A5: **Cosine similarity of memory representations between different serial positions.** We show the cosine similarity between the memory representation of different serial positions, averaged across all trials in the VSR task. We consider the LSTM-256 model for the above plot.

## A4.3 Visualizing Neural Correlates in Task Switching

In the STS experiment, we first take the hidden representations at each $t_{probe}$ for all the test trials within the task and then perform t-SNE [58] to project these hidden representations to 2D.

Next, we introduce two types of color schemes to label hidden representations. First, on the left panel in **Fig. 5A**, we color-code the points based on the ground truth at that particular $t_{probe}$. We assign a blue color to label those hidden representations where the ground truth is 0 i.e. when the marker is either at the left or top of the grid and we assign an orange color to label those hidden representations where the ground truth is 1 i.e. when the marker is either at the right or bottom of the grids. In essence, we color-code the points based on the ground truth response for that particular $t_{probe}$.

Second, on the right panel in **Fig. 5A**, we color-code the hidden representations based on the task that was supposed to be performed at $t_{probe}$. We assign a green color to label those hidden representations when the task at hand is to discriminate top versus bottom and we assign a red color to label those hidden representations when the task at hand is to discriminate left versus right.

By color-coding the neural representations based on the task to be performed at $t_{probe}$, we observed the emergence of two distinct clusters corresponding to the two tasks involved, namely top-versus-bottom and left-versus-right. This observation highlights how the model's neural representations encode and maintain task identity over time, enabling accurate performance in task-switching experiments. Importantly, it should be noted that at $t_{probe}$, the model does not receive any explicit information regarding the task to be performed at that specific time step.

## A4.4 Visualizing Task-Specialized Neural Clusters

Task Variance (TV) [63] is a scalar that quantifies the selectivity of a hidden unit within a recurrent model towards a specific task, where higher values indicate greater selectivity. To obtain the results

in **Fig. 5B, C** in the main text, we first compute the TV for each hidden unit for each task, resulting in a TV matrix of size 10×256 for LSTM-256. We normalize the task variance of each hidden unit such that the maximum normalized TV across all tasks was 1. We then perform t-SNE along the TV dimensions and project the TV vector of each hidden unit to 2D. Subsequently, we identify 8 clusters among the hidden units and assign unique colors to each cluster (**Fig 5B**). In line with [63], we chose the optimal number of clusters based on the highest silhouette score. We then group the hidden units based on their clusters. The sorted TV matrix based on the clusters is shown in **Fig. 5C** in the main text, where the intensity denotes the magnitude of normalized TV values.

Given a pair of tasks A and B, the fractional task variance (FTV) [63] defines the preference of a hidden unit for one task over the other. For a particular hidden unit $i$ in a WM model, the FTV with respect to task A and task B is defined as $FTV_i(A, B) = \frac{TV_i(A) - TV_i(B)}{TV_i(A) + TV_i(B)}$, where $TV_i(A)$ and $TV_i(B)$ are the TVs for tasks A and B, respectively. FTV(A, B) ranges from -1 to 1, with 1 being more selective to task A over B, and vice versa. FTV = 0, when the hidden unit is neutral about both tasks and stays equally active during both tasks. To obtain the results in **Fig. 5D** in the main text, for each pair of tasks, we compute the FTV for each hidden unit and plot the histogram of the total number of hidden units based on their FTV values. In the histogram, the x-axis denotes the FTV whereas the y-axis denotes the proportion of hidden units. If the histogram distribution is skewed towards FTV = 1, it means that the majority of the hidden units are more selective to Task A and vice versa if the distribution is skewed toward FTV = −1. If the distribution is unimodal in the center, it implies that the majority of the hidden units do not have task preference and are equally active in both tasks. In the case of the bimodal distribution where the two modes are located at both ends, this implies that there are two separate populations of neurons, dedicated to two individual tasks respectively.

### A4.5   Causal Analysis of Neural Population

In **Fig A6**, we show the drop in performance for different tasks, when specific neural clusters from the network (shown in **Fig. 5C**) are lesioned. To lesion a neural cluster, we select neurons within that cluster and zero out their projection weights to task-specific classifiers. As anticipated, we note the corresponding causal evidence when distinct clusters are lesioned. For instance, when cluster 6 is deactivated, we observe the maximum decline in performance for the VSR and VSRec tasks. Similarly, when cluster 3 is lesioned, the maximum drop in performance is for the CD task.

### A4.6   Visualizing Task Similarity Matrix

We also visualize the task similarity matrix based on the learned task embeddings for each WM model of different memory capacities in **Fig. A13**. As expected, we observe the brightest values along

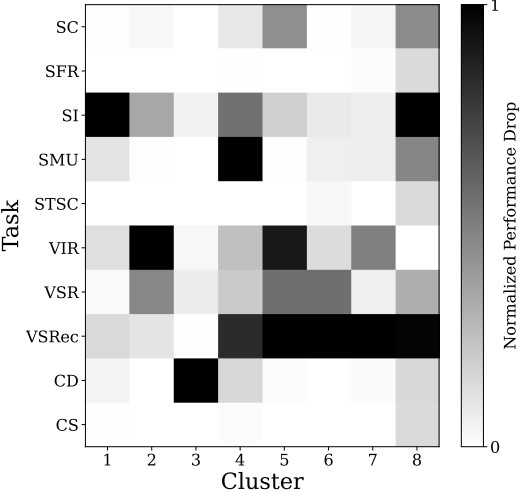

Figure A6: **Effect of lesioning neural clusters on task performance.** The y-axis shows the different tasks and the x-axis shows the neural cluster which is lesioned. See **Sec. 5** and **Fig. 5** in the main text for neural clusters and analysis.

the diagonal of the task similarity matrix as the embeddings from one task compare against itself. Interestingly, we found that in recurrent architectures, task similarities between different conditions within the CD task are high **Fig. A13A-I**. However, we did not observe such strong similarities in transformers **Fig. A13J-L**.

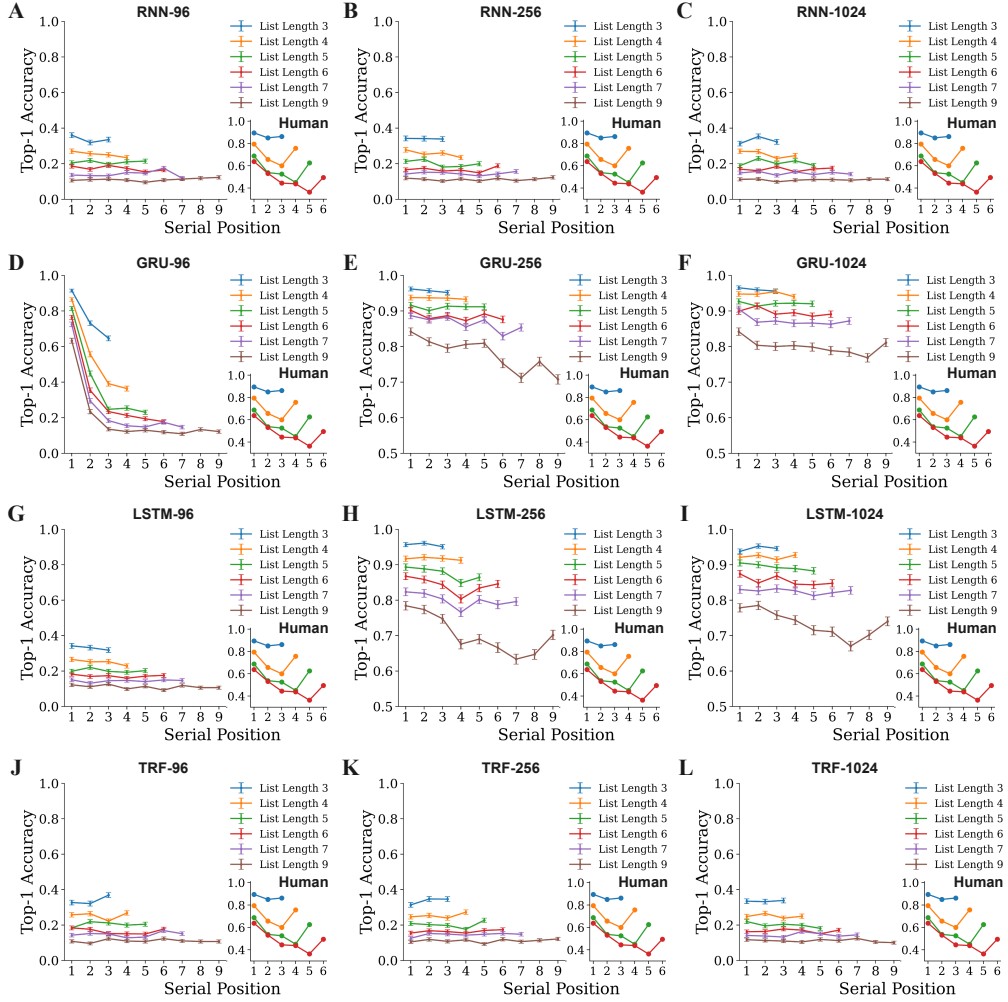

Figure A7: **Behavioral accuracy as a function of list lengths in VSR task.** In addition to **Fig. 4A** in the main text, we show the results of three recurrent architectures (first 3 rows) with three memory capacities (3 columns) and the transformer architecture with two memory capacities (J-L). The chance level is 0.11 for all the above plots.

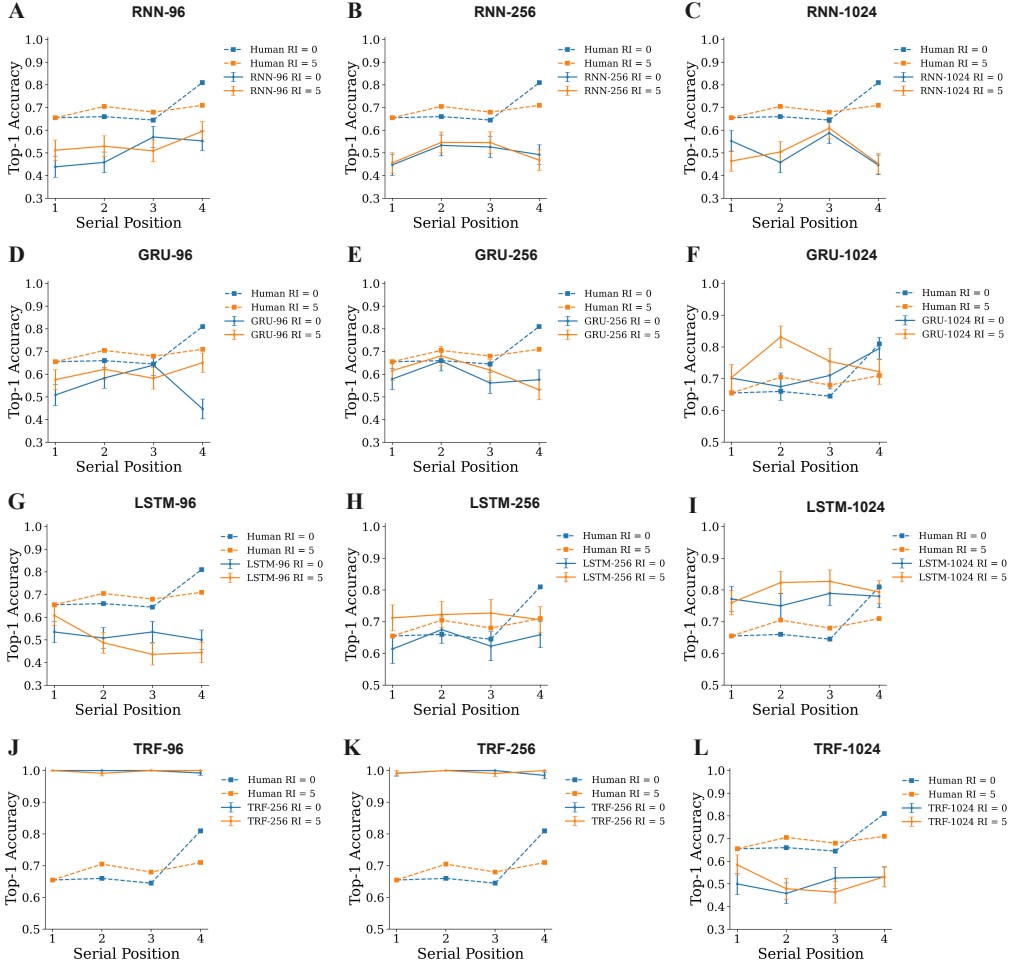

Figure A8: **Behavioral accuracy as a function of retention interval in VIR task.** In addition to **Fig. 4B** in the main text, we show the results of four architectures with different memory capacities. The layout interpretations follow **Fig. A7**. The chance level is 0.5 for all the above plots.

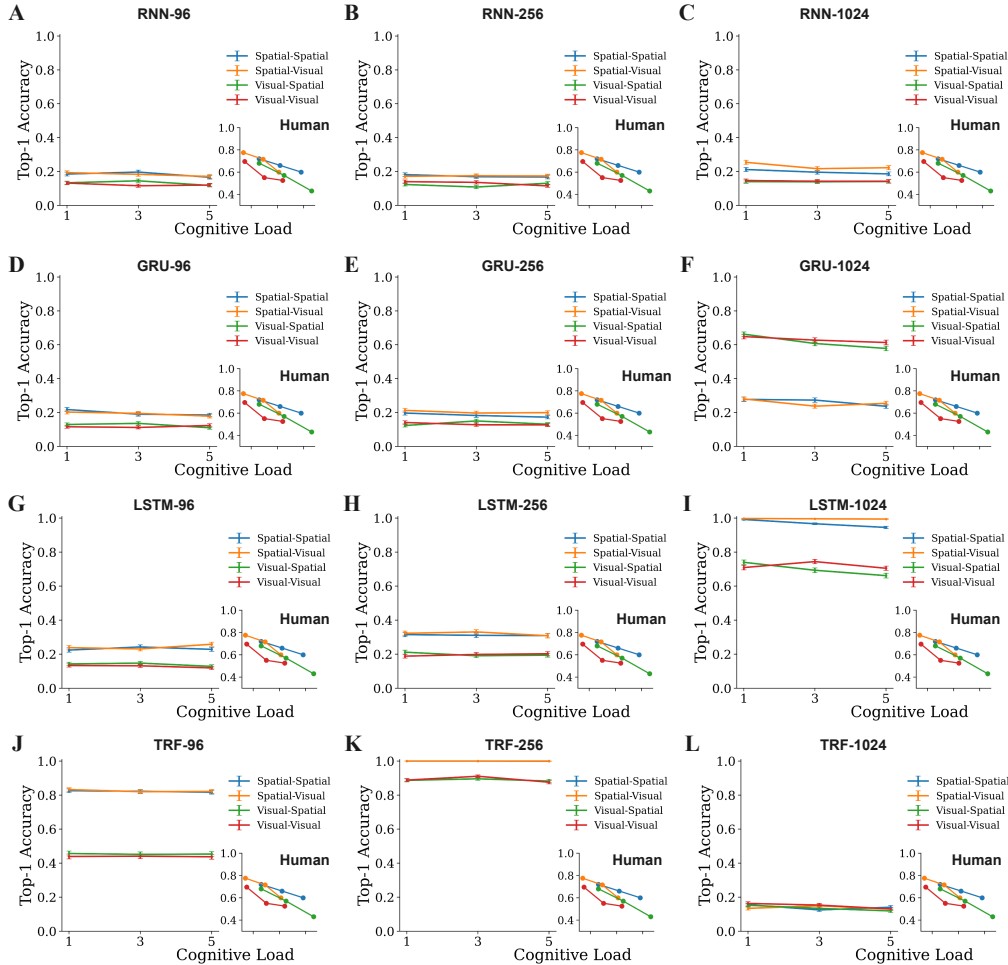

Figure A9: **Behavioral accuracy as a function of memory domain conflicts in CS task.** In addition to **Fig. 4C** in the main text, we show the results of four architectures with different memory capacities. The layout interpretations follow **Fig. A7**. The chance level is 0.04 for all the above plots.

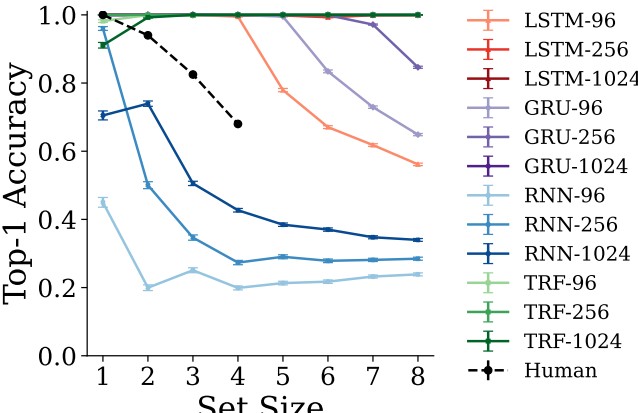

Figure A10: **Behavioral accuracy as a function of set size in SMU task.** In addition to **Fig. 4D** in the main text, we show the results of four architectures with different memory capacities. The layout interpretations follow **Fig. A7**. The chance level is 0.11 for the above plot.

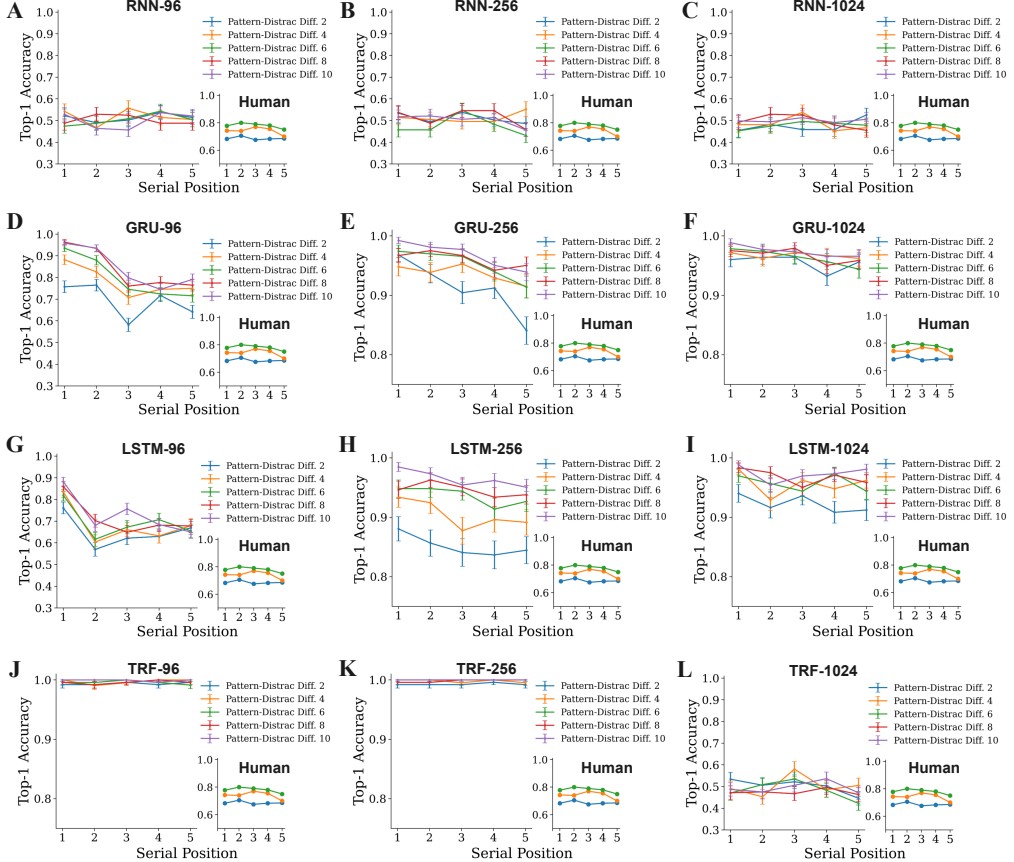

Figure A11: **Behavioral accuracy as a function of memory resolution** $n$ **in VSRec task.** In addition to **Fig. 4E** in the main text, we show the results of four architectures with different memory capacities. The layout interpretations follow **Fig. A7**. The chance level is 0.5 for all the above plots.

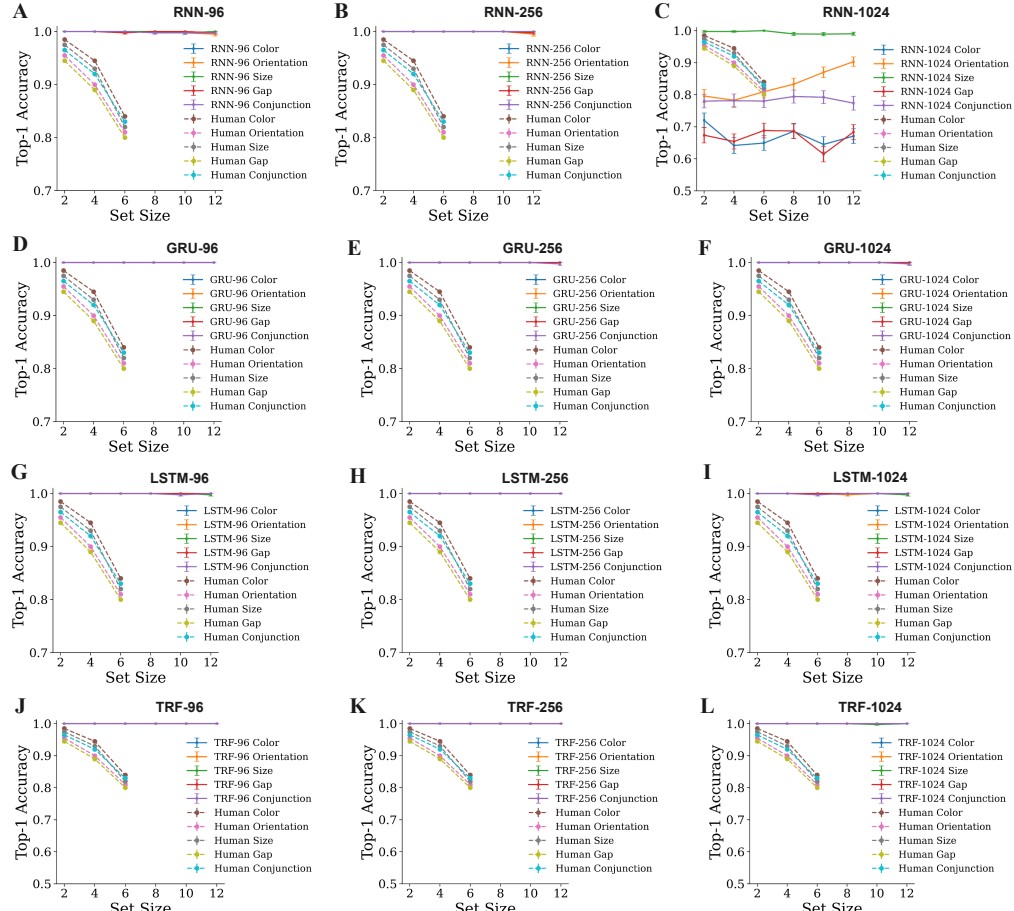

Figure A12: **Behavioral accuracy as a function of features and conjunctions in CD task.** In addition to **Fig. 4F** in the main text, we show the results of four architectures with different memory capacities. The layout interpretations follow **Fig. A7**. The chance level is 0.5 for all the above plots.

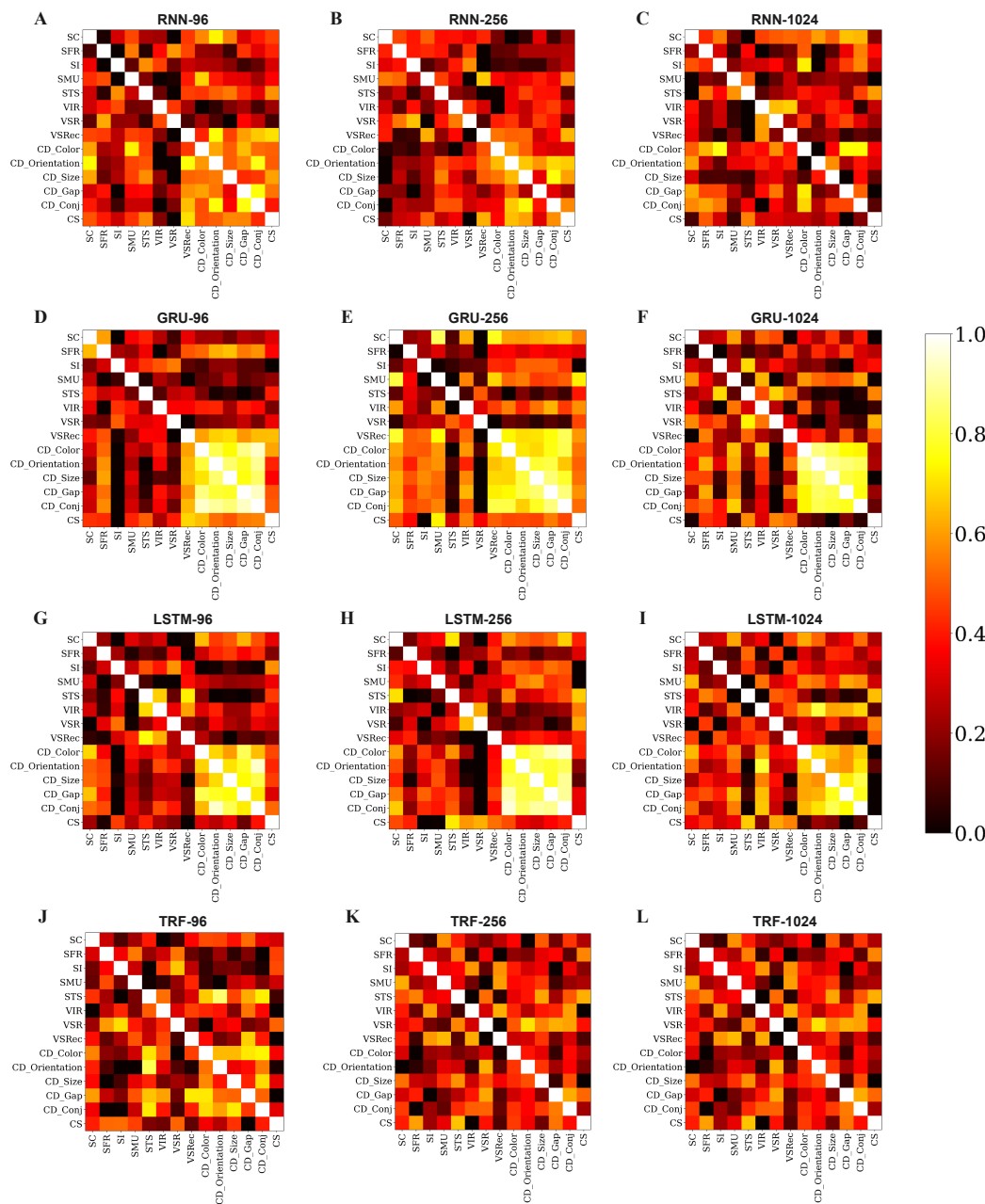

Figure A13: **Visualization of similarity matrix based on learned task embeddings from WM models with different memory capacities.** We take the task embedding learned by each WM model with different memory capacities and compute the cosine similarity between a pair of task embedding vectors from two corresponding tasks. We present the similarity matrix between tasks. Each row of the similarity matrix is normalized such that the maximum similarity is 1 and the minimum is 0. See the color bar on the right for the normalized similarity values. The brighter values indicate that the learned task embeddings are more similar for the two corresponding tasks.

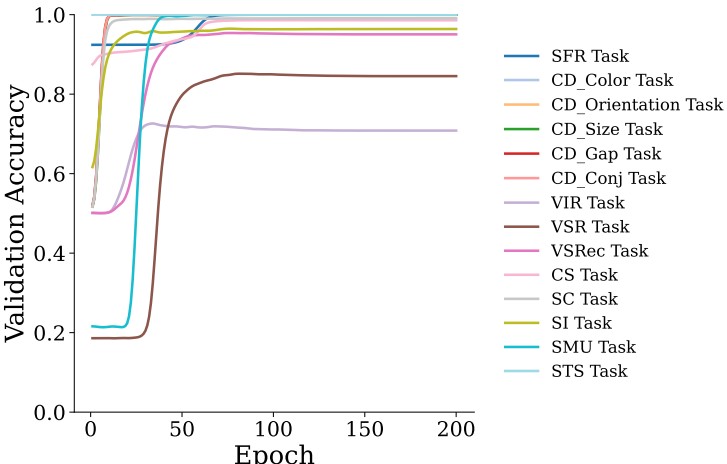

Figure A14: **Training curve of individual tasks for LSTM-1024.** The y-axis shows the validation accuracy for the tasks and the x-axis shows epochs. The training curves for other models show similar trends; thus, they are omitted for simplicity. Here, we plot the model with the highest joint validation accuracy, i.e., LSTM-1024.

## A5 Datasheet for WorM Dataset

### A5.1 Motivation

1. **For what purpose was the dataset created? Was there a specific task in mind? Was there a specific gap that needed to be filled? Please provide a description.**

   Despite significant research progress in studying individual aspects of working memory (WM), there remains a huge gap in the broader study of WM. We take initial steps in this direction by establishing a systematic and quantitative methodology to study the multi-facets of WM. This dataset serves as a valuable resource for communities in cognitive psychology, neuroscience, and AI, offering a standardized framework to compare and enhance WM models, investigate WM's neural underpinnings, and develop WM models with human-like capabilities.

2. **Who created the dataset (e.g., which team, research group) and on behalf of which entity (e.g., company, institution, organization)?**

   The dataset was created by Ankur Sikarwar and Mengmi Zhang from Deep NeuroCognition Lab in affiliation with Center for Frontier AI Research (CFAR), and Institute for Infocomm Research (I2R), from Agency for Science, Technology, and Research (A*STAR), Singapore.

3. **Who funded the creation of the dataset? If there is an associated grant, please provide the name of the grantor and the grant name and number**

   This creation of the dataset is supported by the National Research Foundation, Singapore under its AI Singapore Programme (AISG Award No: AISG2-RP-2021-025) and its NRFF award NRF-NRFF15-2023-0001.

4. **Any other comments?**

   N/A

### A5.2 Composition

1. **What do the instances that comprise the dataset represent (e.g., documents, photos, people, countries)? Are there multiple types of instances (e.g., movies, users, and ratings; people and interactions between them; nodes and edges)? Please provide a description.**

   The dataset consists of 10 distinct working memory tasks. Each data instance in the dataset represents a trial that is time-based and includes stimuli and corresponding responses for various time steps within the trial.

2. **How many instances are there in total (of each type, if appropriate)?**

   The dataset consists of 10 working memory (WM) tasks. One of these tasks i.e. the CD task contains 5 separate task conditions, each of which has separate training, validation, and testing trials. For each of these task conditions within the CD task and 9 remaining WM tasks, there are 86,400 trials for training, 9,600 trials for validation, and 9,600 trials for testing.

3. **Does the dataset contain all possible instances or is it a sample (not necessarily random) of instances from a larger set? If the dataset is a sample, then what is the larger set? Is the sample representative of the larger set (e.g., geographic coverage)? If so, please describe how this representativeness was validated/verified. If it is not representative of the larger set, please describe why not (e.g., to cover a more diverse range of instances, because instances were withheld or unavailable).**

   Since the dataset is synthetic, it is possible to generate an arbitrary number of samples with the provided code. For consistency, we limit the total number of generated samples, including training, validation, and testing, to 105,600 trials for each task.

4. **What data does each instance consist of? "Raw" data (e.g., unprocessed text or images) or features? In either case, please provide a description.**

   Every instance in the dataset includes raw image stimuli corresponding to each time step within the trial, along with the corresponding responses for that particular trial.

5. **Is there a label or target associated with each instance? If so, please provide a description.**

The label for each instance is the ground-truth response corresponding to the task and the specific trial.

6. **Is any information missing from individual instances? If so, please provide a description, explaining why this information is missing (e.g., because it was unavailable). This does not include intentionally removed information, but might include, e.g., redacted text.**

   There is no missing information; the data is complete.

7. **Are relationships between individual instances made explicit (e.g., users' movie ratings, social network links)? If so, please describe how these relationships are made explicit.**

   N/A

8. **Are there recommended data splits (e.g., training, development/validation, testing)? If so, please provide a description of these splits, explaining the rationale behind them.**

   We propose our own training, validation, and testing splits for each task that is available on the GitHub repository.

9. **Are there any errors, sources of noise, or redundancies in the dataset? If so, please provide a description.**

   We generate trials through random sampling. While it is theoretically possible to generate the same trial more than once, the probability of this occurrence is extremely low due to the vast number of possible combinations of stimuli across multiple time steps.

10. **Is the dataset self-contained, or does it link to or otherwise rely on external resources (e.g., websites, tweets, other datasets)? If it links to or relies on external resources, a) are there guarantees that they will exist, and remain constant, over time; b) are there official archival versions of the complete dataset (i.e., including the external resources as they existed at the time the dataset was created); c) are there any restrictions (e.g., licenses, fees) associated with any of the external resources that might apply to a dataset consumer? Please provide descriptions of all external resources and any restrictions associated with them, as well as links or other access points, as appropriate.**

    The dataset is self-contained.

11. **Does the dataset contain data that might be considered confidential (e.g., data that is protected by legal privilege or by doctor–patient confidentiality, data that includes the content of individuals' non-public communications)? If so, please provide a description.**

    N/A. The dataset is synthetic

12. **Does the dataset contain data that, if viewed directly, might be offensive, insulting, threatening, or might otherwise cause anxiety? If so, please describe why.**

    N/A. The dataset is synthetic.

### A5.3 Collection Process

N/A. No data collection process was involved. The dataset has been synthetically generated and we make our dataset generation code available on GitHub. The human data was collected from the existing literature. We directly borrowed their human data and plot them side by side for comparisons with WM models.

### A5.4 Preprocessing/cleaning/labeling

1. **Was any preprocessing/cleaning/labeling of the data done (e.g., discretization or bucketing, tokenization, part-of-speech tagging, SIFT feature extraction, removal of instances, processing of missing values)? If so, please provide a description. If not, you may skip the remaining questions in this section.**

   No preprocessing was done.

2. **Was the "raw" data saved in addition to the preprocessed/cleaned/labeled data (e.g., to support unanticipated future uses)? If so, please provide a link or other access point to the "raw" data.**

   N/A

3. **Is the software that was used to preprocess/clean/label the data available? If so, please provide a link or other access point.**

   N/A

4. **Any other comments?**

   N/A

## A5.5 Uses

1. **Has the dataset been used for any tasks already? If so, please provide a description.**

   The dataset is being used for the first time.

2. **Is there a repository that links to any or all papers or systems that use the dataset? If so, please provide a link or other access point.**

   N/A

3. **What (other) tasks could the dataset be used for?**

   The dataset itself contains ten well-defined WM tasks and is intended to be used for those specific tasks.

4. **Is there anything about the composition of the dataset or the way it was collected and preprocessed/cleaned/labeled that might impact future uses? For example, is there anything that a dataset consumer might need to know to avoid uses that could result in unfair treatment of individuals or groups (e.g., stereotyping, quality of service issues) or other risks or harms (e.g., legal risks, financial harms)? If so, please provide a description. Is there anything a dataset consumer could do to mitigate these risks or harms?**

   No, there are no risks or harms of the dataset. The dataset contains no human data.

5. **Are there tasks for which the dataset should not be used? If so, please provide a description.**

   No.

6. **Any other comments?**

   N/A

## A5.6 Distribution

1. **Will the dataset be distributed to third parties outside of the entity (e.g., company, institution, organization) on behalf of which the dataset was created? If so, please provide a description.**

   The dataset is publicly available.

2. **How will the dataset will be distributed (e.g., tarball on website, API, GitHub)? Does the dataset have a digital object identifier (DOI)?**

   The dataset and the supplementary code is available on the GitHub repository.

3. **When will the dataset be distributed?**

   The dataset is available from June 2023.

4. **Will the dataset be distributed under a copyright or other intellectual property (IP) license, and/or under applicable terms of use (ToU)? If so, please describe this license and/or ToU, and provide a link or other access point to, or otherwise reproduce, any relevant licensing terms or ToU, as well as any fees associated with these restrictions.**

   The dataset is released under the MIT License.

5. **Have any third parties imposed IP-based or other restrictions on the data associated with the instances? If so, please describe these restrictions, and provide a link or other access point to, or otherwise reproduce, any relevant licensing terms, as well as any fees associated with these restrictions.**

   N/A

6. **Do any export controls or other regulatory restrictions apply to the dataset or to individual instances? If so, please describe these restrictions, and provide a link or other access point to, or otherwise reproduce, any supporting documentation.**
   N/A

7. **Any other comments?**
   N/A

## A5.7 Maintenance

1. **Who will be supporting/hosting/maintaining the dataset?**
   Ankur Sikarwar is supporting and maintaining the dataset.

2. **How can the owner/curator/manager of the dataset be contacted (e.g., email address)?**
   ankursikarwardc@gmail.com

3. **Is there an erratum? If so, please provide a link or other access point.**
   The GitHub repository will reflect any changes or improvements made to the dataset.

4. **Will the dataset be updated (e.g., to correct labeling errors, add new instances, delete instances)? If so, please describe how often, by whom, and how updates will be communicated to dataset consumers (e.g., mailing list, GitHub)?**
   This information will be available on GitHub.

5. **If the dataset relates to people, are there applicable limits on the retention of the data associated with the instances (e.g., were the individuals in question told that their data would be retained for a fixed period of time and then deleted)? If so, please describe these limits and explain how they will be enforced.**
   N/A

6. **Will older versions of the dataset continue to be supported/hosted/maintained? If so, please describe how. If not, please describe how its obsolescence will be communicated to dataset consumers.**
   N/A

7. **If others want to extend/augment/build on/contribute to the dataset, is there a mechanism for them to do so? If so, please provide a description. Will these contributions be validated/verified? If so, please describe how. If not, why not? Is there a process for communicating/distributing these contributions to dataset consumers? If so, please provide a description.**
   People who want to contribute are encouraged to get in touch with the authors.

8. **Any other comments?**
   N/A

