# OpenReview forum: "Decoding the Enigma: Benchmarking Humans and AIs on the Many Facets of Working Memory"
_NeurIPS.cc/2023/Track/Datasets_and_Benchmarks — NeurIPS 2023 Datasets and Benchmarks Poster_

### Official Review · Reviewer_E3J7 · 2023-07-18
**Synthetically generated dataset to benchmark ML models' current working memory**

**Rating:** 8
**Confidence:** 4
**Correctness:** Yes

**Strengths:**

- Approach seems novel (although more contextualization is needed)
- Synthetic data generation technique is lucrative
    - Can be used for new tasks as well or to change tasks and generate more data
- Paper results have the potential to be highly insightful and impactful
- Interesting results

Edit:
-This work could be very insightful in testing the flaws of current AI systems when compared to humans as well as providing potential insight into phenomena observed in human working memory.

**Additional Feedback:**

- Should add related works section
- Should develop more experiments that don't allow the model to achieve near 100% accuracy, or make existing experiments more difficult
- Should make clear in main paper that dataset is generated synthetically, should describe briefly in the paper this process

**Clarity:**

The paper is well-written overall. There are some minor issues including:
- In the section where you have Experiment A1-D1 you reference the wrong figure (i.e. 2J instead of 1J)
- Figure 3 not very clear, should just draw a box around all shared parameters
    - Diagram could be much clearer
- Grammar issue(s) abstract
    - "Our results suggest that AI models replicate some characteristics of WM in the brain, most notably primacy and recency effects, and neural clusters and correlates specialized for different domains and functionalities of WM."

**Documentation:**

Yes

**Limitations:**

Limitations of current WM models are briefly discussed. Limitations of the approach/dataset/benchmark are not discussed much - this would be very insightful to readers as to potential routes for further research. Potential negative social impacts of work are not discussed, although I cannot think of any.

**Opportunities For Improvement:**

- Model performance on several tasks was 100% - limits insight - should make tasks more difficult in these cases
- Approach for retention phrase is naive as it is just a gray screen so the model can easily learn to ignore these gray screens
    - Would be nice to see more creative ways of making this more difficult for the model
- Could have used models other than CNN for image encoding, i.e. ViT
- Could have done ablation studies on optimizing each task separately
- Not enough details on data generation
    - I think this is also a way to boost your paper's contribution
- Could do more Short retention interval testing - only did 0 and 5
- Some small issues with linked figures
- I think with some time to fix these things and adding in the related works this paper could be much more impactful and valuable

Edit:
All concerns addressed (those that weren't were due to computational limitations understandably)

**Relation To Prior Work:**

No. Although the paper discusses some past related work, the paper has no related works section which makes it difficult to contextualize its impact with respect to other works. Even if this paper's approach is extremely novel within its field, there are not enough details about past human subject studies as well as research on existing working memory machine learning investigations to properly contextualize this work. I think this is the biggest flaw of this work's current state.

**Summary And Contributions:**

- Generated synthetic dataset
    - Included code to generate new dataset and more samples
    - Included creation of 10 different tasks
- Benchmarked several ML models on dataset
- Compared ML model performance on this WM dataset to human WM performance

---

> ### Author Response · Authors · 2023-08-22
> **Rebuttal by Authors - Part 1**
>
> **We thank the reviewer for the detailed feedback on the paper.**
>
> **E3J7.Improvement.1 - Model performance on several tasks was 100% - limits insight - should make tasks more difficult in these cases**
>
> As the reviewer correctly points out that some models might achieve 100% performance in some working memory tasks. However, there is no single model which performs 100% in all the WM tasks. The reason why some models have 100% accuracy in the CD task is that the model might have sufficiently large memory capacities; hence, it is trivial for the models to remember all the items in the sequence. To tackle this issue, based on reviewer’s suggestion, we added a variation to this task to increase the difficulty. See E3J7.Improvement.2 for more details.
>
> Nonetheless, we would like to emphasize that even when performance of a model on a task is 100%, it can still offer deep insights at the neuronal level explaining how exactly the model solves the task (see Fig 5C in main text and lines 371-376 in supplement for neural analysis and lesion experiments where the LSTM-256 model achieved 100% behaviral accuracy).
>
> **E3J7.Improvement.2 - Approach for retention phrase is naive as it is just a gray screen so the model can easily learn to ignore these gray screens. Would be nice to see more creative ways of making this more difficult for the model**
>
> As the reviewer suggested here, we increased the task difficulty by using random noise images instead of gray screens for every time step in the retention phase in the CD tasks. This also aligns with the backward masking technique in neuroscience, which is often used for disrupting recurrent connections in the biological brains. We added the experiment description and results in line 275-281 in the supplement of the revised version. We trained and tested the LSTM-96 model on this new variant, which was able to completely solve the initial variant of the CD task with 100% accuracy. And indeed, we find that using noisy images during the retention interval makes it difficult for model to overfit on the gray images. Specifically, we observed that LSTM-96 completely fails to learn this new variant of the task.
>
> **E3J7.Improvement.3 - Could have used models other than CNN for image encoding, i.e. ViT**
>
> We would like to highlight that for fair comparisons, all the models including the transformer-based models use the SAME 2D-CNN feature extractors. The goal of our paper is to study the working memory of models, rather than the feature extractors, although we do acknowledge the importance of the feature extractor in the working memory.
>
> As the reviewer suggested, we are now conducting the experiments with various backbones as feature extractors, such as VGG16, SqueezeNet, ViT-small. Note that our model takes a sequence of RGB images as inputs for every trial. It increases the GPU memory usage dramatically for feature extraction backbones with large numbers of network parameters. Due to time constraints and compute resource limits, our experiment is still running. We will update the responses once we have the results. Meanwhile, we will add the results in the final version.
>
> **E3J7.Improvement.4 - Could have done ablation studies on optimizing each task separately**
>
> Following the reviewer’s suggestion, Ideally, we could train and test models on each individual task. However, due to time constraints and compute resource limits, we only managed to train and test our model on the VSR task during the rebuttal period. We will include the rest of the experiments for the remaining 9 tasks in the final version. We provided the experiment description in line 261-262 in the supplement, the results in Fig S4 and the result analysis in line 263-268 in the supplement. The experimental result suggests that the model still exhibits primacy and recency effects. This suggests that the training regimes of the models do not alter our conclusions.
>
>
> **E3J7.Improvement.5 - Not enough details on data generation. I think this is also a way to boost your paper's contribution**
>
> In every experiment, we used computer programs to re-produce the exactly same stimulus set used in the human psychophysics studies in the literature (see Section 5 in main text for the paper reference list) by taking into account various design parameters, such as visual angles in degrees, colors, phase durations, etc. We have provided a detailed documentation in the github for generating such datasets. Moreover, we have made the code customizable and easy to use so that the community can vary its design parameters, including list length, grid sizes, set sizes, etc, to benchmark human and model performances. We believe that our reproduced datasets and our code for generating such datasets are valuable for the community in neuroscience, cognitive science, and computer science. We have now emphasized the design consideration of our WorM dataset in line 175-182 in the main text of the revised version.

---

> > ### Author Response · Authors · 2023-08-22
> > **Rebuttal by Authors - Part 2**
> >
> > **E3J7.Improvement.6 - Could do more Short retention interval testing - only did 0 and 5**
> >
> > We chose retention interval of 0 and 5 by following the designs of human psychophysics experiments. A fixed time scale has been used to convert the retention interval in ms to number of discrete time steps for models (see Table S1 in supplement). As suggested by the reviewer, we now also experimented on retention intervals of 2, 4, and 6. Note that our model takes a sequence of RGB images as inputs for every trial, increasing the list length over a certain limit is intractable with respect to our current GPU memory constraints. Due to time constraints and compute resource limits, we only managed to conduct the experiment above during the rebuttal period. We will include more experiments with longer intervals beyond 5 time steps in the final version. In parallel, to compare with human behaviors, we will also conduct human psychophysics experiments. We added the experiment description in line 270-271 in the supplement of the revised version, presented the experiment result in Fig S5, and added the result analysis in line 271-273 of the supplement. From the experimental results, we did not observe any statistical significance in working memory performance differences across various short retention intervals.
> >
> > **E3J7.Improvement.7 - Some small issues with linked figures**
> >
> > Thank you. We have corrected the wrong figure references.
> >
> > **E3J7.Improvement.8 - I think with some time to fix these things and adding in the related works this paper could be much more impactful and valuable**
> >
> > As suggested by the reviewer, we have added a new Section 2 dedicated to related works in the main text of the revised version. We provided a brief survey of previous research related to modeling working memory and various working memory tasks. However, it is worth noting that these works only study one single individual characteristic of the working memory or test one working memory model on a very limited set of working memory tasks. So far, we have only encountered one study [ref (AjzL) paper] that compiles a battery of working memory tasks comprehensively. However, there was no benchmark of AI models on these tasks. It's important to highlight that our work is the first to propose a large-scale dataset for multiple working memory tasks, on which modern AI models can be trained and tested against humans in various behavioral and neural benchmarks.
> >
> > **E3J7.Limitations.1 - Limitations of current WM models are briefly discussed. Limitations of the approach/dataset/benchmark are not discussed much - this would be very insightful to readers as to potential routes for further research. Potential negative social impacts of work are not discussed, although I cannot think of any.**
> >
> > As the reviewer has alluded to in the previous questions, there are a couple of future directions we can look into in dataset and benchmark designs. First, designing methodology to calibrate various difficulty levels of working memory tasks is interesting, as it gives us ideas about which task is relatively harder for humans/machines to learn and why. Second, there are numerous design variations in each experiment. For example, in the response of **E3J7.Improvement.2**, we can vary the designs of the gray screens during retention period; and hence, modifying the network behaviors. These ablations in the stimulus can provide valuable insights on the working memory mechanisms on humans and AI models. Third, we did not manage to include experiments covering multiple modalities such as audio and language. In the future work, it is interesting to include a set of multi-modal working memory tasks and compare the model/human performances across modalities. Fourth, so far, we have only introduced one training and testing paradigm. To test model generalization abilities, we can try to train the model on one single task and test the model on the remaining tasks. Fifth, we have not exhaustively tested all the state-of-the-art working memory models, especially the biologically-plausible models. It would be interesting to benchmark these models on our WorM dataset. We have included these discussions on future directions in line 344-348 in Section 6 in the revised main text.
> >
> > **E3J7.Clarity.1 - In the section where you have Experiment A1-D1 you reference the wrong figure (i.e. 2J instead of 1J)**
> >
> > Thank you. We have fixed this in the revised version.

---

> > > ### Author Response · Authors · 2023-08-22
> > > **Rebuttal by Authors - Part 3**
> > >
> > > **E3J7.Clarity.2 - Figure 3 not very clear, should just draw a box around all shared parameters. Diagram could be much clearer**
> > >
> > > Thank you. We have revised Figure 3 by adding a bounding box highlighting all the shared parameters.
> > >
> > > **E3J7.Clarity.3 - Grammar issue(s) abstract. "Our results suggest that AI models replicate some characteristics of WM in the brain, most notably primacy and recency effects, and neural clusters and correlates specialized for different domains and functionalities of WM."**
> > >
> > > Thank you. We have now fixed the grammar mistake. See line 10-13 in the abstract of the revised version.
> > >
> > > **E3J7.PriorWork.1 - No. Although the paper discusses some past related work, the paper has no related works section which makes it difficult to contextualize its impact with respect to other works. Even if this paper's approach is extremely novel within its field, there are not enough details about past human subject studies as well as research on existing working memory machine learning investigations to properly contextualize this work. I think this is the biggest flaw of this work's current state.**
> > >
> > > Thank you. We have added a new Section 2 for the related works. See the response of **E3J7.Improvement.8**.
> > >
> > > **E3J7.Feedback -**
> > > **Should add related works section.**
> > > **Should develop more experiments that don't allow the model to achieve near 100% accuracy, or make existing experiments more difficult.**
> > > **Should make clear in main paper that dataset is generated synthetically, should describe briefly in the paper this process.**
> > >
> > > Thank you. See the response of **E3J7.Improvement.1** for increasing the experiment difficulty. See the response of **E3J7.Improvement.5** for elaborating on dataset generation.

---

> > > > ### Comment · Reviewer_E3J7 · 2023-08-27
> > > >
> > > > I really appreciate the amount of effort in addressing reviewer concerns. Almost all of my concerns were addressed and I feel a lot more confident in the potential impact of this work now after seeing the recently added results and edited manuscript. I have increased my score to an 8. I believe this work could be very insightful in testing the flaws of current AI systems when compared to humans as well as providing potential insight into phenomena observed in human working memory.

---

> > > > > ### Author Response · Authors · 2023-08-27
> > > > >
> > > > > We thank the reviewer for the comments. Please let us know if you have other concerns or questions.

---

### Official Review · Reviewer_1qDZ · 2023-07-21

**Rating:** 6
**Confidence:** 3
**Correctness:** The claims made in the paper are corr…

**Strengths:**

- The authors provide a decently large dataset with multiple different tasks having different features, difficulties and capacity to test for different properties of working memory
- The experimental results indicate an important phenomena of alignment of recurrent neural networks to the desirable / observed properties of working memory

**Additional Feedback:**

I do not have any additional feedback.

**Clarity:**

The paper is well written. One thing that is not clear is that the authors claim they provide new evaluation metrics, but the metrics used are primarily Top-1 accuracy for all the tasks. Could the authors provide some clarification on this aspect?

**Documentation:**

The authors provide enough documentation in the framework and the paper, as well as GitHub code to allow for ease of replication.

**Ethics:**

I do not think that there are any ethical concerns with the submission.

**Limitations:**

The primary limitation in the work is that there is no discussion regarding the primacy bias which was observed and its connections with gradient vanishing / exploding property in RNNs. A big problem in RNNs is that distant past is lost over multiple iterations and updates which introduces the question of why the authors are seeing a primacy bias? If you increase the task-lengths, does this bias still exist?

Beyond this, the authors have addressed most limitations and potentially negative societal impacts.

**Opportunities For Improvement:**

- The benchmark proposed by the authors only consider a small temporal length, which biases their observations. A lot of the observations may not hold if training and evaluation were being done at the time-scale of multiple 100 timesteps.
- The tasks are primarily visual-based and it would have been nice to see whether the results demonstrated hold irrespective of the modality (eg. language or audio)

**Relation To Prior Work:**

The connections to prior work is properly made.

**Summary And Contributions:**

The authors propose a dataset and benchmark for understanding the similarities between working memory in humans and the concept of memory in AI systems. Their suite provides them with different kinds of tasks to test for different characteristics that are present in the human working memory. The contribution of this setup is in providing a link to connect how similar recurrent AI systems behave in terms of reasoning abilities and the authors showcase that RNN-style models are more closely aligned with working memory as opposed to transformer styled ones.

---

> ### Author Response · Authors · 2023-08-22
> **Rebuttal by Authors - Part 1**
>
> **We thank the reviewer for their insightful comments and suggestions**
>
> **1qDZ.Improvement.1 - The benchmark proposed by the authors only consider a small temporal length, which biases their observations. A lot of the observations may not hold if training and evaluation were being done at the time-scale of multiple 100 timesteps.**
>
> As the reviewer pointed out, it is interesting to look into longer time sequences beyond 20 time steps. As a preliminary step in this direction, we have now added one more experiment where we ran our model on the VSR task for longer sequences (32 timesteps). Since our model takes a sequence of RGB images as inputs for every trial, a single batch with 100 trials contains 2000 images (20*100) when total number of time steps is 20, so increasing the temporal length over a certain limit is intractable with respect to our current GPU memory constraints. Due to time constraints and compute resource limits, our experiment is still running at the moment. We will update the response once the results are out. Meanwhile, we will include the results of these experiments with longer time sequences in the final version.
>
> Moreover, we would like to emphasize that different behavioral phenomena, like the primacy-receny effect, set-size effect on accuracy, were consistent across different recurrent architectures of varying capacities within the data distribution. In other words, if we train the model on the same data distribution and test the model in the same data distribution, the model’s behavior remains consistent across model variations. For example, in Fig. S6 in supplement, we can observe the primacy and recency effects across recurrent architectures GRU-1024 and LSTM-1024. Moreover, in the same figure, we also observe a similar primacy-recency effect across different capacities of the same architecture, i.e., LSTM-256 and LSTM-1024. Additionally, we show error bars in Fig S6 calculated across trials, to show the consistency of such effects across different test trials.
>
> As a preliminary effort to verify the consistency of the findings, in addition to the experiment on longer time sequences, we also varied the following aspects of some experiments to test whether the original statements made in our paper hold true.
>
> First, we adapted the VSR experiment to language modality task by replacing the visual stimulus with  word embeddings to test whether the primacy and recency effect is a general phenomenon regardless of the modality  (see line 228-239 for experiment description in the supplement, results in Fig. S2, and result analysis in lines 240-245 in the supplement). The results suggest that primacy recency effect is indeed modality-invariant. Also see the response of  **1qDZ.Improvement.2** for more experiment information.
>
> Second, instead of training the model jointly on all the experiments, we now train and test the model only on the VSR task (see line 228-239 for experiment description in the supplement, results in Fig. S2, and result analysis in line 240-245 in the supplement). As expected, we consistently observed the primacy and recency effects across joint training and individual training regimes.
>
> All the above experiments suggest that our behavioral findings seem to be consistent over different training regimes, and multiple modalities.
>
> Last but not least,we have made our code easily customizable and flexible. These customizations include varying list length, trial lengths, set sizes, etc. We have also included detailed documentation on our GitHub repo for better accessibility for users.
>
> **1qDZ.Improvement.2 - The tasks are primarily visual-based and it would have been nice to see whether the results demonstrated hold irrespective of the modality (eg. language or audio)**
>
> As suggested by the review, we now introduce a new experiment based on language. The experiment paradigm follows the one on the Visual Serial Recall experiment. Instead of using visual modality with grid patterns, we instead use language modality, which involves the recall of words. In this task, during the presentation phase, a sequence of words is presented in a particular order. During the recall phase when all the words are presented, the AI models need to choose the words which were presented in the presentation phase in the correct order. We use word2vec to map word tokens into embeddings and the models take these word tokens as inputs at discrete time steps during the presentation phase. We provide the experiment description in line 228-239 of the supplement in the revised version. We present the results in Fig S2 in the supplement and provide the result analysis in line 240-245 in the supplement. From the experimental results, we observed consistent performances as the ones from the Visual Serial Recall experiment. It suggests that the behavioral findings might generalize across different modalities.

---

> > ### Author Response · Authors · 2023-08-22
> > **Rebuttal by Authors - Part 2**
> >
> > **1qDZ.Limitation.1 - The primary limitation in the work is that there is no discussion regarding the primacy bias which was observed and its connections with gradient vanishing / exploding property in RNNs. A big problem in RNNs is that distant past is lost over multiple iterations and updates which introduces the question of why the authors are seeing a primacy bias? If you increase the task-lengths, does this bias still exist?**
> >
> > Indeed! As the reviewer pointed out, we were also surprised and puzzled by the observation that some recurrent neural networks show primacy effects. We attempted the following experiment below.
> >
> > Our hypothesis was that the first stimulus storing in the working memory might act as the first principal component, steering the memory representations of the subsequent stimulus to align with the representation of the first stimulus. By doing so, the first item was forgotten less; thus, the network exhibits the primacy effect. To implement this idea, in every trial of the VSR task, we first compute the hidden states at time step 1, when the first item is presented, and then compute its cosine similarity with the hidden state representations over subsequent time steps t when subsequent memory items are presented. We provided detailed experiment implementation in line 247-256 in Section D, and present the results in Fig S3 and the result analysis in line 256-259 in the supplement. From the results, we observed that cosine similarity of the memory representations at 2nd time step is highest and subsequently decreases monotonically with serial position.
> >
> > We welcome suggestions from the reviewer and we would certainly be happy to conduct more experiments to validate any plausible hypothesis and incorporate these suggestions in the final version.
> >
> > **1qDZ.Clarity.1 - The paper is well written. One thing that is not clear is that the authors claim they provide new evaluation metrics, but the metrics used are primarily Top-1 accuracy for all the tasks. Could the authors provide some clarification on this aspect?**
> >
> > We are sorry for the confusion about the definition of the evaluation metrics. In the old version of the paper, we meant to say that we include all various aspects of working memories that we can evaluate all the computational models on. For example, we introduce primacy and recency effect, set sizes, etc. For each individual aspect, we can calculate a top-1 accuracy score under each condition. Now, we have corrected the use of the term “ evaluation metrics” throughout the paper.
> >
> > Most importantly, to quantitatively evaluate the primacy and recency effects, as well as many other behavior similarity with humans, we now introduce two new types of evaluation scores: slope difference metric and difference in average accuracy metric. We provide math formulation of these scores in line 202-217 in the supplement. Briefly, slope difference metric takes two lines, compute their estimated slopes after linear fitting, and calculate the difference between these two slopes of the model and humans, normalized by the slope of the line for human performances. Average accuracy metric takes the mean accuracy averaged over either serial positions for each condition. With all these evaluation metrics, we report the quantitative measure of behavioral consistency with humans. We present the results in Table S4 and provide the result analysis in line 218-223 in the supplement. Ideally, if the model performs the same as humans, the slope difference would be zero. To test whether the results are significantly different from the chance model, we computed the p-value using two-tailed t-tests. From the results, we found that majority of the models are more consistent with humans than the chance model; however, there still exists performance gaps between these models and humans.

---

> > > ### Author Response · Authors · 2023-08-24
> > > **Additional**
> > >
> > > **1qDZ.Improvement.1 - Additional**
> > >
> > > Following up from the response for **1qDZ.Improvement.1**, now we have completed the training of LSTM-256 model with longer sequence lengths of 32 time steps in the LSR (language serial recall) task. We provided the experiment description in line 386-389 in Sec J in the supplement text, presented the result in Fig S16 and analyzed the results in line 389-393 of the supplement text. From the results, we observe similar primacy and recency effects as we observe in the LSR task with shorter sequence lengths (compare with Fig S2 and line 242-247 in the supplement text for the results on the LSR task when the maximum sequence length is 20). We notice similar trends of primacy effects prominent in shorter list lengths, and the recency effects start emerging in longer list lengths with higher memory load. This result suggests that the primacy-recency effect is a general phenomenon even for longer sequence lengths.

---

> > > > ### Author Response · Authors · 2023-08-28
> > > >
> > > > We thank the reviewer for the suggestions and feedback during the initial review.
> > > >
> > > > We have now provided the responses and the revised manuscript according to the questions and concerns raised by the reviewer. We look forward to engaging in the discussion with the reviewer. We would be happy to clarify any further questions that the reviewer might have.

---

> > > > > ### Comment · Reviewer_1qDZ · 2023-08-28
> > > > > **Official Comment**
> > > > >
> > > > > Thanks to the authors for providing detailed explanations as well as new findings that strengthen their hypothesis. I have increased my score, and would ask the authors to perform an additional experiment if they have time. To understand the primacy bias, the authors computed the cosine similarity at different points with that at the first time-point. However, could the authors instead provide a heatmap of cosine similarities (upper triangular) of every time-point with every other time-point? In particular, their hypothesis holds only if the similarity of the first time-point with the future is more than the same with the second or other time-points with the future.

---

> > > > > > ### Author Response · Authors · 2023-08-30
> > > > > >
> > > > > > We appreciate the reviewer's suggestion regarding the additional experiment. In the final version, we will include the results of the mentioned experiment along with a more detailed analysis of the results.

---

### Official Review · Reviewer_AjzL · 2023-07-22
**Dataset and benchmark for assessing computational WM models**

**Rating:** 4
**Confidence:** 3

**Strengths:**

- Relevant topic: exploring analogies between neural WM implementation and potential artificial counterparts is fruitful direction


**Additional Feedback:**

No additional remarks.

**Clarity:**

The readability is acceptable, although result presentation can be clearer, also discussing possible caveats when undertaking comparison between properties of artificial and natural information processing

**Correctness:**

The presentation has some issues as pointed out above - it is not clear to me whether conclusions on assessing WM capabilities are backed up by the performed experiments.

**Documentation:**

Datasheet for the dataset is provided in the supplementary. Each generated experiment is described in supplementary material. Authors also provide a link to their code for data generation being publically available on github. The documentation on data generation workflow seems to be missing.

**Ethics:**

No ethical concerns given here.

**Limitations:**

Generalization of the obtained results to studies on human WM is in my opinion questionable. Both dataset and employed computational models are limited in their scope, so that it is not clear whether any statements beyond that studied models behave in certain way on the introduced dataset are eligible.

**Opportunities For Improvement:**

- In my opinion, paper lacks proper quantification to allow claimed comparisons. Provided plots make it difficult to understand in what sense human and computational WM performance relate. It would be good if authors make it more clear why comparisons are valid.

- Claims that some architectures (GRU/LSTM) can cope with tasks better than others (Transformers) are hard to check. Are different architectures made to match in compute/param scale used for training so that comparison is valid? Authors should elaborate on this more.

**Relation To Prior Work:**

I think prior work could be highlighted better - what has been already done on modelling side, whether there were already similar benchmarking efforts undertaken. Works like https://psycnet.apa.org/doiLanding?doi=10.1037%2Fbul0000153 suggest that at least efforts to set up systematic benchmarks to compare to computational WM models were existing before - it would be good to have proper overview of what happened so far.

**Summary And Contributions:**

Authors present in their study a synthetically generated dataset containing  10 psychophysics experiments on Working Memory (WM) from cognitive science and neuroscience. To compare mechanisms and phenomena observed in human WM experiments, authors set up a number of computational models of WM by using various types of neural networks, with visual input taken by simple CNNs and further processed by either vanilla RNN, GRU/LSTM, or Transformer backbone. The models are trained on the dataset and their performance is compared to observations from human experiments.


Contributions are as following:

* Generation of WorM, a synthetic WM benchmark dataset, covering 10 WM tasks 60, containing 1 million trials, 4 functionalities and 3 domains of WM.
* Methodology to train and evaluate a selected family of computational WM models on the tasks
* Analysis of 8 WM behavioral benchmarks using the trained computational WM models, including comparison to human WM data and analysis of WM models activations.

---

> ### Author Response · Authors · 2023-08-22
> **Rebuttal by Authors - Part 1**
>
> **We thank the reviewer for their useful feedbacks and suggestions**
>
> **AjzL.Improvement.1 - In my opinion, paper lacks proper quantification to allow claimed comparisons. Provided plots make it difficult to understand in what sense human and computational WM performance relate. It would be good if authors make it more clear why comparisons are valid.**
>
> We report the quantitative measure in top-1 accuracy of all the models for different experiments in Fig 4 in main text and in Fig S6-S11 in the supplement. However, we agree with the reviewer that these top-1 accuracies don’t trivially quantify the consistency between the behavior of AI models and humans. Thus, as suggested by the reviewer, to quantitatively evaluate the primacy and recency effects, as well as many other behavioral similarities with humans, we provide two additional types of evaluation scores: slope difference metric and the difference in average accuracy metric.
>
> We provide math formulation of these scores in section B in line 202-217 in the supplement. Briefly, the slope difference metric takes two lines, compute their estimated slopes after linear fitting, and calculate the difference between these two slopes of the model and humans, normalized by the slope of the line for human performances. Average accuracy metric takes the mean accuracy averaged over either serial positions for each condition. With all these evaluation metrics, we report the quantitative measure of behavioral consistency with humans. We present the results in Table S4 and provide the result analysis in line 218-223 in the supplement. Ideally, if the model performed the same as humans, the slope difference would be zero. To test whether the results are significantly different from the chance model, we computed the p-value using two-tailed t-tests. From the results, we found that majority of the models are more consistent with humans than the chance model; however, there still exists performance gaps between these models and humans.
>
> **AjzL.Improvement.2 - Claims that some architectures (GRU/LSTM) can cope with tasks better than others (Transformers) are hard to check. Are different architectures made to match in compute/param scale used for training so that comparison is valid? Authors should elaborate on this more.**
>
> As the reviewer suggested, we conducted two types of controlled tests: first, we control the working memory capacity of the different models to be of the same size. We defined the working memory capacity in line 243 in the main text of the revised version. Second, we ensured that the total number of network parameters of different models to be of similar size. We provide the description of these two types of controlled experiments in lines 358-370 in the revised supplement. We also list out their memory capacity and their numbers of network parameters in Table S3 in the supplement.
> In Fig. S13, we show the averaged validation accuracy of different models over all WM tasks for easier comparison. For the models with the same memory capacity, we observe that LSTM-1024 achieves the best averaged accuracy over all the tasks, followed by GRU-1024, TRF-1024, and RNN-1024 respectively. Similarly, for the models with the same number of parameters, we can compare RNN-256, GRU-256, LSTM-256, and TRF-96, all of which contain around 2.5 million parameters (shown in Tab S3). Here, based on the averaged accuracy over all tasks, we again observe that LSTM-256 is the best performing model, followed by GRU-256, TRF-96, and RNN-256 respectively. In either comparison, we note that recurrent networks like LSTMs and GRUs perform better than transformers.

---

> > ### Author Response · Authors · 2023-08-22
> > **Rebuttal by Authors - Part 2**
> >
> > **AjzL.Limitations.1 - Generalization of the obtained results to studies on human WM is in my opinion questionable. Both dataset and employed computational models are limited in their scope, so that it is not clear whether any statements beyond that studied models behave in certain way on the introduced dataset are eligible.**
> >
> > **Response Part 1** - We agree with the reviewer that caution is needed when comparing the absolute performances/behavioral phenomena between WM models and humans due to multiple reasons (1) the notions of continuous time in seconds for humans versus discrete time steps for AI models; (2) the learning rules, such as Hebbian learning versus back-propagation; (3) the neuronal mechanistic differences between artificial and biological neurons; (4) the visual diets fed to humans and AI models; and (5) the memory capacity constraints for humans and AI models. However, we would like to point out that all the models in our paper were never trained with any human data or behavioral biases. Yet, these models still remarkably exhibit interesting alignments with human behaviors, such as in dependency on set sizes, pattern similarities, and primacy and recency effects. We now included these discussions in line 355-363 in the main text.
> > Ideally, it would be fantastic to study computational models and perform direct comparisons with biological brains in real-world settings. However, since the birth of neuroscience, neuroscientists have studied underlying mechanisms of cognitive functions in biological brains in well-controlled in-lab experimental settings. For example, in visual attention, search asymmetry is a general phenomenon in humans where searching for target A among distractors B is often easier than searching for target B among distractors A. Researchers [2] discovered that computational models of visual search with CNNs exhibit search asymmetry by testing the visual search models on psychophysics experiments originally tested on humans. Another example is in neuroscience, where [1] used computational models trained on multiple cognitive tasks in a continual learning setting to compare them with prefrontal neurons in monkeys. All these works suggest that computational models are useful tools to approximate the neural correlates and behaviors of biological species. In our work, we follow a similar methodology, utilizing a battery of carefully controlled in-lab psychophysics studies to unveil behaviors exhibited by our models, that were originally identified in humans.
> >
> > In our study, we would like to emphasize that different behavioral phenomena, like the primacy-receny effect, set-size effect on accuracy, were consistent across different recurrent architectures of varying capacities. For example, in Fig. S6 in supplement, we can observe the primacy and recency effects across the recurrent architectures GRU-1024 and LSTM-1024. Moreover, in the same figure, we also observe a similar primacy-recency effect across different capacities within the same architecture, i.e., LSTM-256 and LSTM-1024. Additionally, we show error bars calculated across trials, to show the consistency of such effects across different test trials.
> > As a preliminary effort in this direction, we also varied the following aspects of some experiments to test whether the original statements made in our paper hold true.
> >
> > First, we modify the VSR experiment by replacing the visual stimulus with the word embedding to test whether the primacy and recency effect is a general phenomenon regardless of the modality (see lines 228-239 for experiment description in the supplement, and result analysis in line 240-245 in the supplement). The results suggest that primacy recency effect is modality-invariant.
> >
> > Second, instead of training the model jointly on all the experiments, we now train and test the model only on the VSR experiment (see line 261-268 in supplement for experiment description and results). As expected, we consistently observed primacy and recency effects across joint training and individual training regimes.
> >
> > All the above experiments suggest that the statements made in our paper are consistent over different training regimes, and multiple modalities.

---

> > > ### Author Response · Authors · 2023-08-22
> > > **Rebuttal by Authors - Part 3**
> > >
> > > **AjzL.Limitations.1 - Generalization of the obtained results to studies on human WM is in my opinion questionable. Both dataset and employed computational models are limited in their scope, so that it is not clear whether any statements beyond that studied models behave in certain way on the introduced dataset are eligible.**
> > >
> > > **Response Part 2** - The reviewer’s comments are also closely related to the out-of-distribution generalization problem in the machine learning literature. Since the introduction of ImageNet, the entire community has started developing large vision models, that can perform well on ImageNet. Later studies show that most of these models still fail to generalize in out-of-domain distributions, such as real-world images [3]. Up to today, we still recognize the significant contribution of ImageNet and many other image datasets; although we agree that solving ImageNet does not solve ALL the computer vision problems, including out-of-distribution generalization.
> > >
> > > Similarly, in our paper, we did NOT make any claims about the out-of-distribution generalization ability of the current working memory models we tested. Instead, we emphasized the following contributions. We introduce the large working memory dataset (WorM) and set up a framework to evaluate current and even future working memory models. We list several working memory characteristics and introduce evaluation metrics on these aspects. Moreover, we introduce several example baseline models and test them on our WorM dataset. Via the demonstration of how the current working memory models perform in our WorM dataset, our study reveals several interesting insights.
> > >
> > > Moreover, our WorM dataset provides an ideal arena to benchmark the out-of-distribution generalization ability in working memory tasks for AI models. Among the many use cases of our WorM dataset, the community can train the models on limited number of time steps and test whether the models can generalize in longer time sequences. The community can also train the models on a subset of tasks and test the models on the remaining subset of tasks. We have made our code implementations easy-to-use and convenient-to-customize for such generalization studies on humans and AI models. In line 344-348 in the main text of the revised version, we emphasized that one of the important use of our WorM dataset is to benchmark the generalization ability of the working memory models.
> > >
> > > References:
> > >
> > > [1] Yang, G. R., Joglekar, M. R., Song, H. F., Newsome, W. T., & Wang, X. J. (2019). Task representations in neural networks trained to perform many cognitive tasks. Nature neuroscience, 2022.
> > >
> > > [2] Gupta, S. K., Zhang, M., Wu, C. C., Wolfe, J., & Kreiman, G. (2021). Visual search asymmetry: Deep nets and humans share similar inherent biases. NeurIPS 2021.
> > >
> > > [3] Barbu, A., Mayo, D., Alverio, J., Luo, W., Wang, C., Gutfreund, D., ... & Katz, B. (2019). Objectnet: A large-scale bias-controlled dataset for pushing the limits of object recognition models. NeurIPS 2021.
> > >
> > >
> > >
> > > **AjzL.Correctness.1 - The presentation has some issues as pointed out above - it is not clear to me whether conclusions on assessing WM capabilities are backed up by the performed experiments**
> > >
> > >
> > > We have now addressed the issue on assessing WM capabilities. See the response of **AjzL.Improvement.2**.
> > >
> > >
> > > **AjzL.Clarity.1 - The readability is acceptable, although result presentation can be clearer, also discussing possible caveats when undertaking comparison between properties of artificial and natural information processing**
> > >
> > >
> > > As the reviewer suggested, we discussed the differences between artificial and biological information processes in line 355-363 of Section 6 in the main text. To briefly summarize, artificial neurons in neural networks operate by generating "firing rates," whereas biological neurons introduce additional temporal dimensions through processes like frequency and phase conversion, resulting in brainwave bands like alpha and beta. These brainwave bands are crucial in functions such as memory encoding, consolidation, and replays, especially within the hippocampus. Furthermore, there are disparities in how time is perceived and utilized by humans and AI models. AI models function with a sequence of discrete time steps, whereas humans conduct experiments within a continuous temporal framework. Another point of contrast is the visual input received by humans versus models. Humans are exposed to a continuous stream of visual stimuli throughout the year, whereas AI models are trained specifically on synthetic images designed for working memory tasks. Lastly, current deep recurrent neural networks acquire knowledge through backpropagation, while biological systems more frequently rely on biologically plausible learning mechanisms such as Hebbian learning. Looking ahead, it would be intriguing to assess the performance of more biologically-inspired working memory models, like attractor networks, using the WorM dataset.

---

> > > > ### Author Response · Authors · 2023-08-22
> > > > **Rebuttal by Authors - Part 4**
> > > >
> > > > **AjzL.PriorWork.1 - I think prior work could be highlighted better - what has been already done on modelling side, whether there were already similar benchmarking efforts undertaken. Works like https://psycnet.apa.org/doiLanding?doi=10.1037%2Fbul0000153 suggest that at least efforts to set up systematic benchmarks to compare to computational WM models were existing before - it would be good to have proper overview of what happened so far.**
> > > >
> > > > We thank the reviewer for referring to us to this fantastic working memory benchmark paper. We loved reading this paper! The paper presents a battery of interesting and representative working memory benchmarks. We cited this paper and discussed this work in lines 76-80 in Section 2 in the main text.  We highlighted that our work included benchmarks of computational models of working memories and publicly available code for implementing these models, which were originally missing in the review paper. Nevertheless, we found that the benchmark paper is super exciting. In the future work, we plan to incorporate several additional working memory tasks, such as Benchmark 6 on auditory effect and Benchmark 7 on word lengths from the benchmark paper into our WorM dataset. These two additional experiments include working memory tests in language and audio domains, which will increase the modality diversity in our WorM dataset. As the reviewer suggested, we also added the entire Section 2 on related works and expanded the surveys and discussions of the follow-up works on the benchmark paper.
> > > >
> > > > **AjzL.Documentation.1 - Datasheet for the dataset is provided in the supplementary. Each generated experiment is described in supplementary material. Authors also provide a link to their code for data generation being publically available on github. The documentation on data generation workflow seems to be missing.**
> > > >
> > > > We thank the reviewer for pointing this out. We have now provided more documentation on our Github repository about the data generation workflow. This documentation now includes notebooks showing how the stimulus looks like for each experiment, which script to run in order to generate the stimulus for different tasks, and which parameters are available for customizing the stimulus designs and trial conditions. We have also added notebooks where users can visualize the WM tasks with different experiment conditions, evaluate them, and conduct behavioral and neural analysis.

---

> ### Author Response · Authors · 2023-08-28
>
> We thank the reviewer for the suggestions and feedback during the initial review.
>
> We have now provided the responses and the revised manuscript according to the questions and concerns raised by the reviewer. We look forward to engaging in the discussion with the reviewer. We would be happy to clarify any further questions that the reviewer might have.

---

### Official Review · Reviewer_9dCi · 2023-07-23

**Rating:** 7
**Confidence:** 5
**Correctness:** They are relatively appropriate and s…
**Clarity:** Yes

**Strengths:**

- The authors study DL models from the perspective of working memory, which is a critical concept of understanding human intelligence. It is a critical topic from the perspective of both DL and neuroscience that deserves more attention from the community.
- The authors provide a large-scale and carefully curated dataset, with several DL models trained and human psychophysics experiments conducted. This provide a solid and high-quality contribution.
- The authors provide a diverse set of experiments that systematically evaluate different perspective of WM of models.
- The paper is nicely written.

**Additional Feedback:**

See above

**Documentation:**

The code is available. I would suggest the authors include more details in their repo's Readme, and also provide more examples and notebooks to guide the readers to use the proposed dataset and to conduct analysis.

**Limitations:**

The authors discussed limitations.

**Opportunities For Improvement:**

- The experiments seem very short in terms of its length. I wonder if the authors have thought about how the experiments could be conducted, and how the results would be altered, if the tasks contain a much longer sequence?

- In Figure 4, human seems to perform worse than many models, which violate the authors' claim that human performance could be an upper bound of the model's performance. I think this is related to how the concept of time differ in terms of model v.s. human, and how the experiments are conducted/designed for models under such constraint. The authors could include more discussion/analysis towards this.

- The authors should include more details about the human psychophysics experiments in the main text.

- The authors could include more hyper parameter selection details.

Minor comments and questions:
- line 29. What is the meaning of "investigating the individual differences in WM abilities?"
- line 68. The notations of different phases are slightly confusing and might be improved if possible.
- line 104-152. I believe the experiments should refer to Figure 1 instead of Figure 2. Please correct the in-line reference accordingly.

**Relation To Prior Work:**

The authors could discuss more related works.

**Summary And Contributions:**

This work aims to establish a framework to understand how deep learning networks can be used to study working memory (WM). In order to do so, the authors provide a dataset that consists of 1 million different trials with 10 WM tasks, which contains many perspectives of functionalities, domains, and neural characteristics. To evaluate how the dataset can be used, the authors trained RNNs and transformers to investigate how such models would behave in terms of executing memory functions. The authors further provided human psychophysics experiments to provide an upper bound for the models' performance. The authors further study the models from a neural population perspective to build connections between human v.s. deep learning models, providing insights on building better bio-inspired artificial intelligence systems.

---

> ### Author Response · Authors · 2023-08-22
> **Rebuttal by Authors - Part 1**
>
> **We appreciate the comments, suggestions and improvements proposed by the reviewer**
>
> **9dCi.Improvement.1 - The experiments seem very short in terms of its length. I wonder if the authors have thought about how the experiments could be conducted, and how the results would be altered, if the tasks contain a much longer sequence?**
>
> We have used the maximum number of time steps as 20 for all the experiments. We have made our code easily customizable and flexible, if future studies require longer time sequences. These customizations also include varying list lengths, trial lengths, set sizes, etc. We have also included detailed documentation on our GitHub repo for better accessibility for users.
>
> As the reviewer pointed out, it is interesting to look into longer time sequences beyond 20 time steps. As a preliminary step in this direction, we have now added one more experiment where we ran our model on the VSR task for longer sequences (32 timesteps). Since our model takes a sequence of RGB images as inputs for every trial, a single batch with 100 trials contains 2000 images (20*100), so increasing the list length over a certain limit is intractable with respect to our current GPU memory constraints. Due to time constraints and compute resource limits, our experiment is still running at the moment. We will update the response once the results are out. Meanwhile, we will include the results of these experiments with longer time sequences in the final version.
>
> **9dCi.Improvement.2 - In Figure 4, human seems to perform worse than many models, which violate the authors' claim that human performance could be an upper bound of the model's performance. I think this is related to how the concept of time differ in terms of model v.s. human, and how the experiments are conducted/designed for models under such constraint. The authors could include more discussion/analysis towards this.**
>
> Yes, the reviewer is right! Humans do not get to be the upper bound by solely comparing the absolute performances between humans and models reported in Figure 4. We have now removed the phrases “upper bound” and modified the statement in Line 10 and Line 43 in the main text of the revised manuscript.
>
> Indeed, as correctly pointed out by the reviewer, the absolute performance difference between humans and models should not be over-interpreted. There could be a couple of reasons for such performance gaps. One could be the notions of time and time steps used for humans and AI models. The other reason could be the difference in visual diets between humans and models. While humans have been “trained” on 365-day videos a year; AI models are specifically trained on synthetic images used for working memory tasks. However, we would like to highlight that the purpose of our paper is to compare behavioral consistency between AI models and humans, such as whether both models and humans exhibit primacy and recency effects, rather than the absolute performance differences between AI models and humans.
>
> In many cases, we would still like to argue that humans remain superior to AI models in the working memory tasks. For example, although humans have never been specifically trained on any of these working memory tasks, they are amazing at performing these working memory tasks in a zero-shot manner. Clearly, this is not the case for all the models we tested here, as they have to be trained on thousands of examples in the working memory tasks. Second, humans are very good at generalizing beyond a fixed set of time steps/set sizes/retention intervals. The generalization of AI models beyond what they have been trained on remains one of the challenges in AI. From these perspectives, humans still have superior working memory capabilities over AI models. We have included these discussion points in lines 355-363 in Section 6 in the main text.
>
> **9dCi.Improvement.3 - The authors should include more details about the human psychophysics experiments in the main text.**
>
> We would like to clarify that we did NOT conduct human psychophysics experiments ourselves. We directly use the human data collected from the previous works in the psychology/neuroscience literature (see Section 3 for each experiment’s reference) and re-produce their results in our paper for easy comparison. We added this statement in line 113 in the main text of the revised version to avoid further confusion.
> We would appreciate the reviewer to point out exactly what more experiment details should be included in Section 3 in the main text. We would be happy to add these details in the final version.

---

> > ### Author Response · Authors · 2023-08-22
> > **Rebuttal by Authors - Part 2**
> >
> > **9dCi.Improvement.4 - The authors could include more hyper parameter selection details.**
> >
> > As the reviewer suggested, we have included more details on hyperparameter selection in lines 190-191 in in the supplement. Hyperparameter tuning was done using grid search. We have also included Table S2 in the revised supplement, showing the different hyperparameters we explored. We would be obliged to add more details in the final version if the reviewer could pinpoint exactly what hyper-parameters are missing.
> >
> > **9dCi.Questions.1 - line 29 What is the meaning of "investigating the individual differences in WM abilities?"**
> >
> > By “investigating the individual differences in WM abilities”, we were referring to studies that investigated how WM abilities differed in people of different ages, neurological disorders, and intelligence levels. We have rephrased this in line 28-29 in the main text of the revised version as follows:
> > ”..and (3) investigating how WM abilities differed in individuals of different ages [19 , 49], intelligence levels [12], and neurological disorders [2, 21].”
> >
> > **9dCi.Questions.2 - line 68 The notations of different phases are slightly confusing and might be improved if possible.**
> >
> > We would really appreciate it if the reviewer could provide more concrete suggestions about which phase notations are confusing. We would be happy to improve the text in the final version.
> >
> > **9dCi.Question.3 - line 104-152 I believe the experiments should refer to Figure 1 instead of Figure 2. Please correct the in-line reference accordingly.**
> >
> > Thank you! We have now corrected the figure references in line 119 - 170 in the main text of the revised version.
> >
> > **9dCi.Documentation - The code is available. I would suggest the authors include more details in their repo's Readme, and also provide more examples and notebooks to guide the readers to use the proposed dataset and to conduct analysis**
> >
> > As suggested by the reviewer, we have included more details in the Readme about how to jointly train and test models on our WorM dataset. We have also provided pretrained models of different architectures and capacities for easy reproducibility. Finally, we added detailed instructions on how one can generate their own data for different tasks with custom trial conditions, as well as multiple notebooks for easy data visualization, behavioral and neural analysis. (“viz_data.ipynb”, "bench_vsrec_task.ipynb" "bench_cd_task.ipynb" "bench_cs_task.ipynb" "bench_neural_analysis.ipynb" "bench_smu_task.ipynb" "bench_sts_task.ipynb" "bench_vir_task.ipynb" "bench_vsr_task.ipynb")

---

> > ### Author Response · Authors · 2023-08-24
> > **Additional**
> >
> > **9dCi.Improvement.1 - Additional**
> >
> > Following up from the response for **9dCi.Improvement.1**, now we have completed the training of LSTM-256 model with longer sequence lengths of 32 time steps in the LSR (language serial recall) task. We provided the experiment description in line 386-389 in Sec J in the supplement text, presented the result in Fig S16 and analyzed the results in line 389-393 of the supplement text. From the results, we observe similar primacy and recency effects as we observe in the LSR task with shorter sequence lengths (compare with Fig S2 and line 242-247 in the supplement text for the results on the LSR task when the maximum sequence length is 20). We notice similar trends of primacy effects prominent in shorter list lengths, and the recency effects start emerging in longer list lengths with higher memory load. This result suggests that the primacy-recency effect is a general phenomenon even for longer sequence lengths.

---

> ### Author Response · Authors · 2023-08-28
>
> We thank the reviewer for the suggestions and feedback during the initial review.
>
> We have now provided the responses and the revised manuscript according to the questions and concerns raised by the reviewer. We look forward to engaging in the discussion with the reviewer. We would be happy to clarify any further questions that the reviewer might have.

---

> > ### Comment · Reviewer_9dCi · 2023-08-28
> > **Thanks for the revision and responses**
> >
> > Thank you for the careful responses provided by the authors.
> > - Thank you for the careful comments and new results for Improvement.1, this has strengthened the contribution of the paper.
> > - Thank you for the clarification for Improvement.2 and Improvement.3, I think the new discussion points are of high quality.
> > - Thank you for the Improvement.4, I think the newly added details made the work more reproducible and rigorous.
> > - Thank you for addressing Questions.1, I think the new version improved the readability of the work.
> > - For Questions.2, I was referring to notations as $P_{present}$ etc could be improved if possible, but I do not have a concrete plan for this and also it is a relatively minor comment. I just got personally confused by this and think if there are any better notations it could be clearer.
> > - Thank you for your edition towards the Documentation, I haven't got the time to go through the new codebase yet but I appreciate the efforts to improve it.
> >
> > Overall I think that is a good and acceptable paper in the intersection of neuroscience and ML. I agree with reviewer Unmh that the paper tries to address a key gap and the authors made substantial efforts towards it, and I personally as a researcher would be happy to play with their dataset and thus I think it would be great if the paper is accepted.

---

> > > ### Author Response · Authors · 2023-08-30
> > >
> > > **9dCi.FurtherQuestions - For Questions.2, I was referring to notations as $P_{present}$ etc could be improved if possible, but I do not have a concrete plan for this and also it is a relatively minor comment. I just got personally confused by this and think if there are any better notations it could be clearer.**
> > >
> > > We thank the reviewer for the general comments about our work. Indeed, we hope that the publication of our dataset can benefit the neuroscience and machine learning community in advancing the development of the working memory models in artificial intelligence systems and understanding the working memory mechanisms in biological systems.
> > >
> > > We also thank the reviewer for the clarification on Question 2. In the final version, to increase the readability of the notations for different phases, we will elaborate individual phase notations in the text by grounding them in concrete example experiments and adding these notations in the legends of Figure 1 and Figure S1.

---

### Official Review · Reviewer_Unmh · 2023-07-27
**Decoding the Enigma Review**

**Rating:** 8
**Confidence:** 2
**Correctness:** Yes
**Clarity:** Yes

**Strengths:**

* The authors' effort to bridge the gap between neuroscience, cognitive science, and AI is significant. They identify a key gap in research – the lack of integrative, systematic study and benchmarking across multiple WM tasks, and take substantial steps to address this through the introduction of the comprehensive, large-scale WM benchmark dataset.
* The authors introduce a comprehensive set of 10 WM tasks and use them to evaluate state-of-the-art recurrent neural networks and transformer models
* The models produce key effects from the human data and classic working memory literature, including primacy effects and recency effects
* The paper clearly outlines its methodology, experiments, and provides detailed explanations of the tasks involved in the study. This transparency enhances the reproducibility of the research.
* The "neural" analysis of the models is compelling and illustrates the utility of looking into the network to interpret its dynamics

**Additional Feedback:**

N/A

**Documentation:**

Yes

**Limitations:**

Yes

**Opportunities For Improvement:**

*  There seem to be no statistical tests conducted to confirm effects such as the primacy and recency effects, which are fundamental in understanding the behavior and dynamics of memory models and human behavior. I suggest they back up any claims with the proper statistical tests that would be expected for human experiments
* Given that they have full control over the deep neural networks they train (as opposed to the human brain), it would be interesting to augment the neural analysis with ablations or artificial stimulation to gain causal evidence about the functions of the neural populations they analyze
* Relatedly, recency effects in an LSTM make sense, but primacy effects less though. It would be useful for authors to dig into this further and potentially even offer a mechanism for this

**Relation To Prior Work:**

Yes

**Summary And Contributions:**

The authors present a robust and comprehensive contribution to both neuroscience and AI research, making strides in advancing our understanding of working memory (WM) and modeling this cognitive function within artificial intelligence systems. They introduced an exhaustive benchmark dataset composed of 10 tasks, incorporating over a million trials, which spanned 4 functionalities and 3 domains of WM. In addition, they put both WM models and humans to the test on these 10 tasks, comparing their 11 memory attributes at the behavioral, performance, and neural stages using a new set of evaluation metrics. They garnered understanding regarding the foundational workings of WM through these tasks and pinpointed the deficiencies in current WM models.

---

> ### Author Response · Authors · 2023-08-22
> **Rebuttal by Authors**
>
> **We thank the reviewer for taking the time to review our paper and to provide helpful feedback**
>
> **Unmh.Improvement.1 - There seem to be no statistical tests conducted to confirm effects such as the primacy and recency effects, which are fundamental in understanding the behavior and dynamics of memory models and human behavior. I suggest they back up any claims with the proper statistical tests that would be expected for human experiments**
>
> We agree with the reviewer. We now compute the error bars (denoting standard errors) over all the test trials for different models and update all the plots in Fig. 4 and Figs. S6-S11 in the main text and supplement.
>
> Moreover, to quantitatively evaluate the primacy and recency effects, as well as other behavioral similarities with humans, we provide two types of evaluation scores: slope difference metric and the difference in average accuracy metric. We provide the math formulation of these scores in lines 202-217 in Section B of the revised supplement. Briefly, the slope difference metric takes two lines, computes their estimated slopes after linear fitting, and calculates the difference between these two slopes of the model and humans, normalized by the slope of the line for human performances. The average accuracy metric takes the mean accuracy averaged over the serial positions for each condition. With all these evaluation metrics, we report the quantitative measure of behavioral consistency of models with humans. We present the results in Table S4 and provide the result analysis in line 236-240 in the main text. Ideally, if the model performed the same as humans, the slope difference would be zero. To test whether the resutls are significantly different from the chance model, we computed the p-value using two-tailed t-tests. From the results, we found that the majority of the models are more consistent with humans than the chance model; however, there still exists performance gaps between these models and humans.
>
>
> **Unmh.Improvement.2 - Given that they have full control over the deep neural networks they train (as opposed to the human brain), it would be interesting to augment the neural analysis with ablations or artificial stimulation to gain causal evidence about the functions of the neural populations they analyze**
>
> The suggestion from the reviewer is very interesting! We followed the reviewer’s suggestion and knocked out the neurons corresponding to their cluster numbers, as shown in Fig. 5c in the main text, to investigate causal evidence. To achieve this, we first select the neurons in a particular neural cluster, and then zero out their projection weights to all the task-specific classifiers. We described the implementation steps in lines 371-373 in the supplement, presented the results in Fig. S12, and discussed the results in lines 373-376 in the supplement. As expected, we observe the corresponding performance drops in the accuracy of specific tasks depending on the neural cluster that has been lesioned. For example, in Fig. 5c in the main text, the 3rd neural cluster is responsible for the CD experiment. After lesioning this cluster, we saw a decrease in Top-1 accuracy from 0.99 to 0.65. Our experimental results demonstrate the causal evidence of different populations involved in solving different sets of tasks.
>
>
> **Unmh.Improvement.3 - Relatedly, recency effects in an LSTM make sense, but primacy effects less though. It would be useful for authors to dig into this further and potentially even offer a mechanism for this**
>
> Indeed! As the reviewer pointed out, we were also surprised and puzzled by the observation that LSTM shows primacy effects. We attempted the following experiment below.
>
> Our hypothesis was that the first stimulus stored in the working memory might act as the first principal component, steering the memory representations of the subsequent stimulus to align with the representation of the first stimulus. By doing so, the first item was forgotten less; thus, the network exhibits the primacy effect. To implement this idea, in every trial of the VSR task, we take the hidden states at time step 1 as the reference vector, and then compute its cosine similarity with the hidden state representations over subsequent time steps t. We provided implementation details in line 247-256 in Section D of the supplement, present the results in Fig S3 and the result analysis in line 256-259 in the supplement. From the results, we observed a monotonically decreasing trend in cosine similarity over all time steps. This indicates that the first item in the working memory acts as an important reference when subsequent hidden states align with the representation of the first item, leading to less forgetting.
>
> We welcome suggestions about other possible explanations from the reviewer and we would certainly be happy to conduct more experiments to validate any plausible hypothesis and incorporate these suggestions in the final version.

---

> ### Author Response · Authors · 2023-08-28
>
> We thank the reviewer for the suggestions and feedback during the initial review.
>
> We have now provided the responses and the revised manuscript according to the questions and concerns raised by the reviewer. We look forward to engaging in the discussion with the reviewer. We would be happy to clarify any further questions that the reviewer might have.

---

> > ### Comment · Reviewer_Unmh · 2023-08-28
> > **Response**
> >
> > Nice work implementing these statistics and new analyses! The causal evidence results and primacy effect result are really interesting. I have raised my score to an 8

---

> > > ### Author Response · Authors · 2023-08-29
> > >
> > > We thank the reviewer for the comments. Please let us know if you have other concerns or questions.

---

### Author Response · Authors · 2023-08-22
**General Comment to all the Reviewers**

We thank all the reviewers for the feedback. Below, we provide responses to individual questions from different reviewers. We also submitted a revised manuscript (both the main text and the supplement) where the added content is highlighted in BLUE. All the figures, tables, and line numbers referenced in the rebuttal are based on the revised manuscript.

---

### Decision · Program_Chairs · 2023-09-22

**Decision:**

Accept (Poster)

**Comment:**

The paper receives mostly positive reviews except one. The positive reviews acknowledges the size and insightful prelimiary empirical studies with the proposed dataset. Although the one negative reviewer raised seemingly valid questions, the authors provide comprehensive answers to them and the reviewer did not actively respond to the authors' response. Given the value of the proposed dataset, the AC recommends to accept the submission to NeurIPS 23 DB track.